# Two failure modes of deep transformers and how to avoid them: a unified theory of signal propagation at initialisation

**Alessio Giorlandino & Sebastian Goldt**
International School of Advanced Studies (SISSA)
Trieste, Italy
{agiorlan, sgoldt}@sissa.it

## Abstract

Finding the right initialisation for neural networks is crucial to ensure smooth training and good performance. In transformers, the wrong initialisation can lead to one of two failure modes of self-attention layers: rank collapse, where all tokens collapse into similar representations, and entropy collapse, where highly concentrated attention scores lead to training instability. While previous work has studied different scaling regimes for transformers, an asymptotically exact, down-to-the constant prescription for how to initialise transformers has so far been lacking. Here, we provide an analytical theory of signal propagation through deep transformers with self-attention, layer normalisation, skip connections and MLP. Our theory yields a simple algorithm to compute trainability diagrams that identify the correct choice of initialisation hyper-parameters for a given architecture. We overcome the key challenge, an exact treatment of the self-attention layer, by establishing a formal parallel with the Random Energy Model from statistical physics. We also analyse gradients in the backward path and determine the regime where gradients vanish at initialisation. We demonstrate the versatility of our framework through three case studies. Our theoretical framework gives a unified perspective on the two failure modes of self-attention and gives quantitative predictions on the scale of both weights and residual connections that guarantee smooth training.

## 1 Introduction

Finding the right initialisation for a neural network is a key challenge facing every practitioner before training. Choosing the right scale for the initial weights is critical for maintaining information flow during the forward and backward passes, which in turn ensures smooth training and good final performance. In fully connected networks, a series of works established a clear picture of signal propagation and trainability at initialisation by computing how the similarity of two inputs evolves as they propagate through the same network at initialisation (Poole et al., 2016; Schoenholz et al., 2016; Xiao et al., 2018; Hanin and Rolnick, 2018; Hanin, 2018; Doshi et al., 2023), ultimately establishing the optimal initialisation at the "edge of chaos" (Poole et al., 2016).

In Transformers (Vaswani et al., 2017), where fully-connected layers alternate with self-attention layers (Bahdanau et al., 2015), the key quantity for measuring information flow through the network is the similarity between tokens in a sequence as it propagates through the network. Signal propagation faces additional challenges due to two key failure modes of self-attention layers. The first is *rank collapse*, where self-attention maps all input tokens to identical representations, producing an output matrix of rank one. This phenomenon manifests as the attention pattern shown in fig. 1(a). Dong et al. (2021) demonstrated that networks composed solely of self-attention layers will inevitably collapse any input sequence into uniform token representations at a rate that is double exponential in the number of layers. This rank collapse fundamentally destroys input sequence information and impedes effective training by inducing vanishing gradients (Noci et al., 2022). *Entropy collapse* represents the second failure mode, where queries attend to only a small, frozen subset of tokens irrespective of the input, leading to low Shannon entropy of the attention distribution (hence the name) and, more

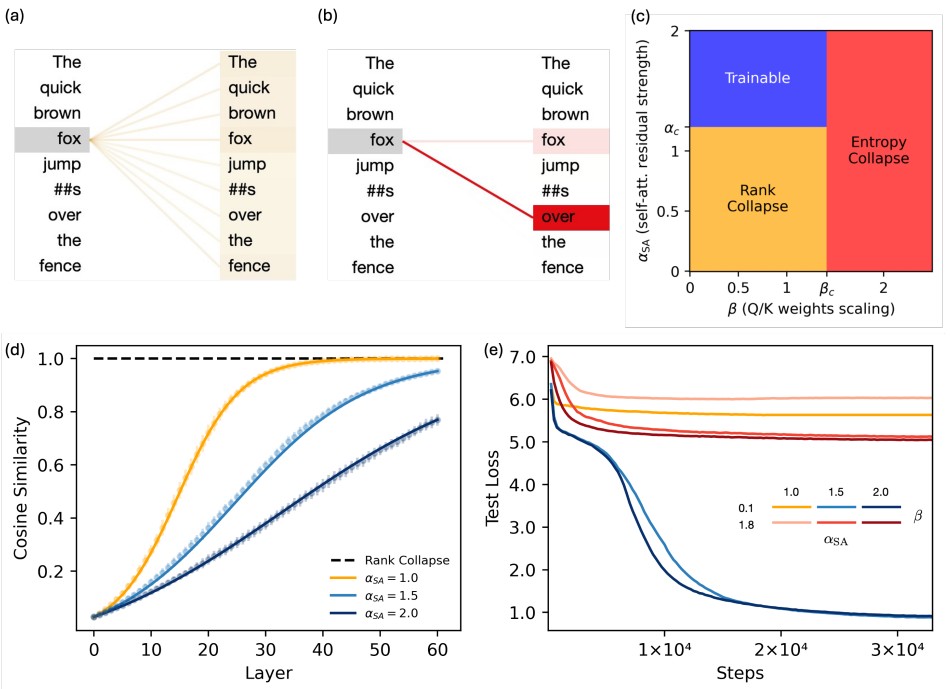

Figure 1: **Two failure modes of Transformers at initialisation, and how to avoid them. (a)** Rank collapse occurs when the self-attention layer attends uniformly to all tokens, mapping all input tokens into the same output token. **(b)** Entropy collapse is a regime of highly saturated attention matrices which attend to random, semantically meaningless patterns, leading to training instability (Zhai et al., 2023). **(c)** Trainability diagram for a 60-layer BERT Transformer, obtained from our analytical theory of signal propagation, see algorithm 1. Depending on the strength of the self-attention residual connections $\alpha_{\text{SA}}$ (eq. (5)) and the scale of initial key and query weights $\beta$ (eq. (9)), we delineate the three regimes of rank collapse, entropy collapse, and the regime where the Transformer is trainable (blue). **(d)** Average cosine similarity between token embeddings of a sequence taken from the TinyStories dataset as it propagates through the layers of a vanilla BERT model for different self-attention residual strengths; empirical measurements (dots) closely follow theoretical predictions (solid lines). Sufficiently large residual connections $\alpha_{\text{SA}}$ are key to preventing the similarity between tokens from becoming unity, which would indicate rank collapse. **(e)** Test loss of a 60-layer BERT model on TinyStories for two initialisations from each regime. Models suffering from rank or entropy collapse at initialisation fail to train, as predicted by theory. Full experimental details in section A.1.

importantly, unstable training (Zhai et al., 2023). An attention matrix exhibiting this pathology is shown in fig. 1(b).

## 1.1 RELATED WORK

Previous work has highlighted the importance of skip connections in mitigating rank collapse (Dong et al., 2021; Noci et al., 2022; Wang et al., 2024), and Zhai et al. (2023) suggested a modification of the self-attention layer that helps avoiding entropy collapse. However, a quantitative, unified description of how these two distinct phenomena emerge has been lacking. Our result 1 fills this gap, providing clear guidance on how to jointly avoid both issues by appropriately choosing the strength of residual connections and the scale of initial weights.

The main challenge in studying information propagation in Transformers arises from the self-attention layers. Previous works either assume uniform attention (Noci et al., 2022) or approximate the average behaviour of a self-attention layer by taking expectations separately over the numerator and denominator of the softmax (Cowsik et al., 2024) – a strong simplification that fails the behaviour of large initial query and key weights, which is responsible for entropy collapse. In contrast, we analyse the standard softmax Transformer in the complementary limit of infinitely long sequences

by leveraging tools and concepts from statistical physics. Our theoretical framework provides a unified explanation for – and a practical solution to – the emergence of both failure modes observed in practice: rank collapse and entropy collapse.

**Further related works.** Geshkovski et al. (2023a;b) developed an elegant mathematical framework that models Transformers as interacting particle systems, revealing emergent clustering phenomena in token representations. Boncoraglio et al. (2025) determines Bayes-optimal generalisation error for attention-indexed models at finite sequence length $T$, while we consider the complementary limit of $T \to \infty$. Bordelon et al. (2024) analyses mean-field training *dynamics* in various limits: infinite query/key dimensions, infinite heads, and infinite depth, leveraging dynamical mean field theory to derive feature and gradient kernels. Noci et al. (2023) provide a description of the limiting distribution of the covariance matrix of token embeddings in the proportional limit of infinite width and depth for a modified Transformer architecture while we explore the complementary limit of infinite sequence length. We revisit their result in section 5.

## 1.2 CONTRIBUTIONS

**Our main contribution** is an analytical theory of signal propagation in deep Transformers with self-attention layers, skip connections, layer normalisation, and MLPs. Our theoretical framework yields a simple algorithm to compute the evolution of the typical overlap between token embeddings as sequences propagate through deep, off-the-shelf Transformers at initialisation. This enables us to construct trainability diagrams such as the one shown in fig. 1(c) for a 60-layer BERT-style Transformer. By varying the scale of initial key and query weights $\beta$, eq. (9), and the strength of the skip connections of the self-attention layer $\alpha_{\mathrm{SA}}$, eq. (5), we identify three regimes for signal propagation: entropy collapse dominance (red), rank collapse dominance (yellow), and a trainable regime characterised by small initial weights and strong skip connections (blue). The critical threshold for query/key variance, $\beta_c$, emerges as a global property, while the critical threshold $\alpha_c$ for residual strength depends on network depth: for any given model depth, our theory predicts the minimum residual strength required to guarantee signal propagation and ensure trainability.

**We validate our theory in two ways.** In fig. 1(d), we show that our theory (solid lines) accurately predicts the average cosine similarity between tokens (averaged over all token pairs in a sequence) when propagating sequences from the TinyStories dataset (Eldan and Li, 2023) through a vanilla BERT model (Devlin et al., 2019) at initialisation (dots). In fig. 1(e), we show the test loss of a 60-layer BERT model on TinyStories for two initialisations from each regime; models suffering rank or entropy collapse at initialisation indeed fail to train, as the trainability diagram predicts. We provide further applications of our theory in section 5.

We also make the following **additional contributions**:

- We identify a sharp phase transition in the behaviour of self-attention in the limit of infinite sequence length, using tools from statistical physics (Derrida, 1981). Below a critical value $\beta_c$ of the initial scale $\beta$ of query/key weights, eq. (9), attention matrices are approximately uniform, while for $\beta > \beta_c$, they are highly saturated, causing rank and entropy collapse, respectively, see result 1.

- Our results are exact in the limit of very large sequences. We calculate finite-size corrections that are essential for obtaining precise, practical predictions, see section 3.2.

- We analyse the backward pass by deriving an exact expression for the norm of the gradient of query and key weights at initialisation, and find that gradients vanish for $\beta < \beta_c$, see section 3.3.

- We integrate our results on self-attention and normalisation layers with existing results on MLPs to derive simple algorithms like algorithm 1 that yield trainability diagrams like the one shown in fig. 1c, see section 4.

- We demonstrate the versatility of our framework by considering three case studies: signal propagation in a standard BERT architecture; comparing different placements of LayerNorm; and comparing variations of the self-attention mechanism itself, see section 5.

In summary, we provide a full theory of signal propagation in deep Transformers that unifies two phenomena previously discussed separately, namely entropy and rank collapse. Tuning initialisation

hyperparameters in accordance with our theoretical analysis enables accurate prediction of the trainability of very deep models, offering practitioners a principled approach to designing and initialising Transformer architectures for reliable training at scale.

## 2 SETUP AND NOTATION

We consider a vanilla Transformer encoder (Vaswani et al., 2017) that processes sequences $\{X_t\}_{t=1,\ldots,T}$ of $T$ tokens embedded in a $d$-dimensional space. We analyse signal propagation through a complete Transformer block comprising self-attention, residual connections, layer normalisation, and a feed-forward MLP.

**Layer Norm.** We consider layer normalisation (Ba et al., 2016) which centers each token embedding and rescales it by its standard deviation; for simplicity, we omit the affine transformation:

$$\mathrm{LN}(X_t) = \frac{X_t}{\sqrt{\mathrm{Var}[X_t]}} = \frac{X_t}{d^{-1/2}\|X_t\|}, \tag{1}$$

where $\|X_t\|$ is the Euclidean norm. We consider both pre-norm and post-norm variants.

**Self-Attention.** Given the query, key and value weight matrices $W_Q, W_K, W_V \in \mathbb{R}^{d \times d}$, the attention score between tokens $X_t$ and $X_{t'}$ is

$$a_{tt'} = \frac{1}{\sqrt{d}} (W_Q X_t)^\top (W_K X_{t'}). \tag{2}$$

These scores are normalised via a softmax operation to obtain the attention weights, which are then used to compute the weighted sum of the value projections:

$$A_{tt'} = \frac{e^{a_{tt'}}}{\sum_{\tau=1}^{T} e^{a_{t\tau}}}, \qquad \mathcal{S}(X)_t = \sum_{t'=1}^{T} A_{tt'} W_V X_{t'}. \tag{3}$$

**MLP.** Each embedding $X_t$ is processed independently by a shared two-layer feed-forward network with hidden dimension $d$ and non-linearity $\phi$:

$$\mathrm{MLP}(X_t) = W_2\, \phi(W_1 X_t + b_1) + b_2, \tag{4}$$

The weight matrices $W_1, W_2$ and the biases $b_1, b_2$ are initialised with i.i.d. entries of zero mean and variance $\sigma_w^2/d$ and $\sigma_b^2$, respectively.

**Residual Connections.** We write the residual connections for the self-attention and MLP blocks as

$$\mathrm{RES}_{\mathrm{SA}}(X) = \mathcal{S}(X) + \alpha_{\mathrm{SA}} X, \qquad \mathrm{RES}_{\mathrm{MLP}}(X) = \mathrm{MLP}(X) + \alpha_{\mathrm{MLP}} X, \tag{5}$$

where $\alpha > 0$ are the strengths of the skip connections.

**Sequence Structure.** Since token embeddings are sampled i.i.d. at initialisation, standard concentration results (Vershynin, 2018, sec. 3.1) in high dimension $d$ imply that their norms concentrate, while pairwise overlaps scale as $O(d^{-1/2})$. Under self-attention, these tokens mix and their overlaps evolve as the sequence propagates in depth. Two key quantities to describe the evolution of **token similarity** are the overlap and cosine similarity matrices

$$q_{ts} = \frac{1}{d} X_t \cdot X_s \qquad \rho_{ts} = \frac{q_{ts}}{\sqrt{q_{tt} q_{ss}}}, \tag{6}$$

which measure the degree of alignment between two tokens in a sequence. Our theory provides a framework to analytically track the evolution of these quantities through transformer blocks.

**Notation.** We denote the average over tokens or token pairs with brackets $\langle \cdot \rangle$, whilst $\mathbb{E}[\cdot]$ denotes the average with respect to the initialisation of the parameters $W_Q, W_K, W_V$.

## 3 A THEORY FOR SIGNAL PROPAGATION IN TRANSFORMERS

The main challenge in analysing information flow through a Transformer is handling the self-attention mechanism with its strong non-linearity and its normalisation step. In this section, we derive a theory for signal propagation by leveraging a formal similarity between self-attention and the *Random energy model* of Derrida (1981), see section 3.1. Using tools and concepts from statistical physics, we give an asymptotically precise characterisation of signal propagation in self-attention layers in the forward pass in section 3.2. We analyse the backward pass in section 3.3. To complete the picture, we finally add the remaining elements of the Transformer architecture in section 4, which yields algorithm 1 to compute the trainability diagram of Transformers.

### 3.1 SELF-ATTENTION AND THE RANDOM ENERGY MODEL

The starting point of our analysis is an exact mapping of attention at initialisation to a model of disordered systems from statistical physics, called the Random Energy Model (Derrida, 1981). This mapping was previously identified by Lucibello and Mézard (2024) in the context of associative memories; our novelty lies in leveraging it to provide an exact description of signal propagation.

**The Random Energy Model (REM).** The REM is a simple model of a disordered system with $N$ spins $\mathbf{s}$ that has $\mathcal{O}(\exp(N))$ possible configurations. Each configuration has an energy $E(\mathbf{s})$ drawn at random from a Gaussian distribution with variance $\mathcal{O}(N)$ – hence the name Random Energy Model. The probability of each state is given by the Boltzmann distribution with inverse temperature $\beta > 0$:

$$p(\mathbf{s}) = \frac{e^{-\beta E(\mathbf{s})}}{\mathrm{Tr}_{\mathbf{s}} \, e^{-\beta E(\mathbf{s})}}. \tag{7}$$

The probability of a state $p(\mathbf{s})$ and the attention $A_{tt'}$ between a pair of tokens, eq. (3), thus share the same mathematical structure: a normalised exponential with a random argument. While the energies in the REM are random by construction, the self-attention scores $a_{tt'}$ are random at initialisation due to their dependence on the random initial weights. We therefore interpret a row of the attention matrix as a REM, with the full attention matrix representing $T$ copies of the system.

**Mapping of Self-Attention to the REM.** We initialise the query/key projection parameters i.i.d. with variance $\sigma_Q^2 = \sigma_K^2 = {}^{\sigma_a}/d$ so that the attention scores, defined in eq. (2), are normally distributed with mean 0 and variance $\sigma_a^2$. While both REM energies and self-attention scores $a_{tt'}$ are Gaussian, the key difference between them is that attention scores are correlated. Averaging over the realisation of $W_Q, W_K$, we get the following correlation structure (see section B.1):

$$\mathbb{E}\left[a_{ts} a_{\tau\sigma}\right] = \sigma_Q^2 \sigma_K^2 \left(X_t \cdot X_\tau\right)(X_s \cdot X_\sigma) = \sigma_a^2 q_{t\tau} q_{s\sigma}. \tag{8}$$

We extend the REM framework to account for these correlations; consequently, the critical inverse temperature derived in result 1 depends on these underlying geometric quantities.

The only free parameter is $\sigma_a^2$, and thus the initialisation of query/key weights, which controls the scale of attention scores and, as we will see later, the sharpness of the self-attention mechanism. The remaining step is to match the scaling of attention scores with sequence length to the random energies.

**Scaling of Query/Key Weights.** In MLPs, the natural scaling for weight variance is $\mathcal{O}(1/d)$ (He et al., 2015), ensuring that neuron pre-activations remain $\mathcal{O}(1)$. For self-attention, the REM analogy suggests that the natural scaling for query and key weight variance should make the attention score variance $\mathcal{O}(\log T)$, where $T$ is the sequence length, to match the energy of a REM with $N$ degrees of freedom, whose energy typically scales as $\mathcal{O}(N)$ with fluctuations of $\mathcal{O}(\sqrt{N})$, while the partition sum runs over $\mathcal{O}(e^N)$ configurations. Self-attention only sums over $T$ tokens in the sequence, suggesting that attention scores should scale like $\mathcal{O}(\sqrt{\log T})$. This motivates us to control the variance of attention scores via a constant inverse "temperature" parameter $\beta \in \mathbb{R}$ as:

$$\sigma_Q^2 = \sigma_K^2 = \sigma_a/d \qquad \sigma_a = \beta\sqrt{\log T}. \tag{9}$$

This scaling is, to our knowledge, novel and complements the existing literature on various reparametrisations of the infinite-width limit (Yang and Hu, 2020) by accounting for the infinite sequence length limit.

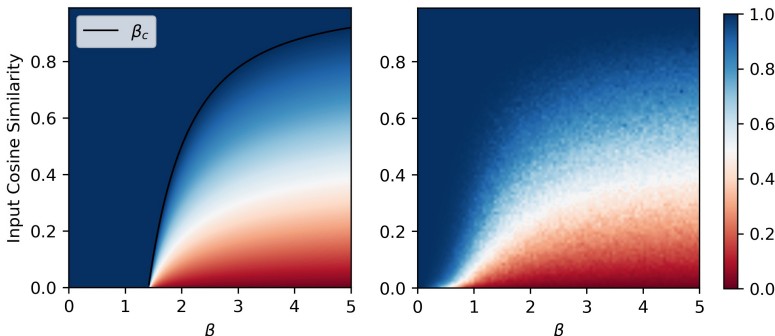

Figure 2: **Phase diagram for a single layer of self-attention.** We use result 1 to plot the average cosine similarity between pairs of tokens after one layer of self-attention as a function of the query/key variance parameter $\beta$ and the input average cosine similarity $\rho$. *(Left)*: Theoretical phase diagram obtained from result 1 (with $q = 1$ and $p = \rho$). For $\beta < \beta_c$, we observe a *rank collapse* phase, where all input tokens map to a single output direction and the cosine similarity saturates at 1. For $\beta > \beta_c$, token diversity is preserved, but *entropy collapse* emerges. *(Right)*: Simulations with embedding dimension $d = 512$ and sequence length $T = 1024$ qualitatively reproduce the theoretical transition, with deviations attributed to finite-size effects, as discussed in section 3.2.

## 3.2 Forward Signal Propagation through a self-attention layer

By tracking the average squared norm $\mathbb{E}\langle q_{tt}\rangle := q$ and the average overlap $\mathbb{E}\langle q_{ts}\rangle := p$, we describe the evolution of the average cosine similarity $\rho$ and of the inverse participation ratio (IPR), which serves as an indicator of entropy collapse. The IPR of an attention row is defined for all queries $t \in T$ and $r \in \mathbb{N}$ as

$$Y_t^{(r)} = \sum_{s=1}^{T} A_{ts}^r. \tag{10}$$

For $r = 2$, the IPR estimates the effective number of tokens receiving attention: a small IPR, scaling as $o_T(1)$, corresponds to delocalised attention, while a non-vanishing IPR indicates that only $O_T(1)$ keys matter, signalling sparse self-attention and entropy collapse. Larger values of $r$ sharpen the distinction between localised and delocalised attention vectors.

We state the result below and provide its derivation in section B.2.

**Result 1** (Average Cosine Similarity Update under Self-Attention). *Let $W_Q$ and $W_K$ be initialised with i.i.d. entries with variance $\sigma_Q^2 = \sigma_K^2 = \beta\sqrt{\log T}/d$, and $W_V$ with variance $\sigma_V^2 = \sigma_v^2/d$. For a sequence with average squared token norm $q$ and average pairwise overlap $p$, define the critical initialisation scale $\beta_c(q, p) \equiv \sqrt{\frac{2}{q(q-p)}}$. In the limit of infinite width $d \to \infty$ and infinite sequence length $T \to \infty$, we then have that:*

*1. The evolution of the average cosine similarity $\rho$ takes the form:*

$$\Phi_{\mathcal{S}}(\rho) = \frac{\rho}{(1-\rho)Y^{(2)}(\beta) + \rho} = \begin{cases} 1, & \beta < \beta_c(q, p), \\ \frac{\rho}{1 - \beta^{-1}\sqrt{2(1-\rho)}}, & \beta > \beta_c(q, p). \end{cases} \tag{11}$$

*2. The average inverse participation ratio (IPR) $Y_t^{(2)}$ satisfies $\forall t \in [T]$:*

$$\lim_{T\to\infty} \mathbb{E}Y_t^{(2)} = Y^{(2)}(\beta) = \begin{cases} 0, & \beta < \beta_c(q, p) \\ 1 - \frac{\beta_c(q,p)}{\beta}, & \beta > \beta_c(q, p) \end{cases} \tag{12}$$

Result 1 allows us to draw a phase diagram that describes the behaviour of self-attention for different values of $\beta$ and the input cosine similarity $\rho$, see fig. 2. Before we discuss the phase diagram in detail, we comment on the definition of the critical $\beta_c$ and on finite-size effects.

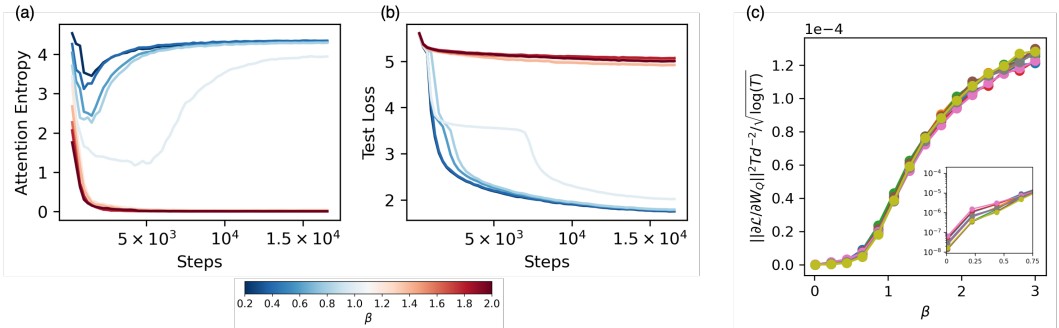

Figure 3: *(a, b)* **A phase transition in the impact of query / key initialisation on training dynamics.** Average Shannon entropy of attention's row and the test loss of a Transformer with a single layer of self-attention trained on masked language modelling on TinyStories as we vary the scale of the initialisation from small to large initial weights (blue to red). Small initial weights (blue) permit attention to diversify over time, supporting effective learning, while large-variance initialisation (red) collapses the attention to only a few tokens, visible in an entropy that quickly goes near zero. Here $\beta_c(\rho = 0) = \sqrt{2}$. *(c)* **Norm of the query gradient.** Frobenius norm of the gradient of the loss with respect to query weights for various combinations of sequence length $T = 2048, 4096, 8192$ and embedding dimension $d = 256, 512, 1024$. As predicted by result 2, gradients collapse for different $T$ and $d$, and vanishing gradients afflict the low-$\beta$ regime.

**Definition of $\beta_c$ and Finite-Size Effects.** A key element of our analysis is the convergence behaviour of the softmax normalisation $Z(\beta) = \sum_{\tau=1}^{T} e^{a_{t\tau}}$, which depends on $\beta$ through the scores $a_{t\tau}$ (see eq. (2)). We define $\beta_c$ as the value of $\beta$ at which the law of large numbers for $Z$ breaks down, i.e. the value for which $Z(\beta)$ no longer concentrates around its mean as $T \to \infty$. However, as noted by Ben Arous et al. (2005), the central limit theorem for $Z$—which guarantees Gaussian fluctuations around $\mathbb{E}Z$—already fails at a lower threshold $\tilde{\beta}_c = \beta_c/2$. Since $T$ is large but finite in practice, we therefore expect our asymptotic predictions to remain accurate for $\beta < \tilde{\beta}_c$, where fluctuations are still approximately Gaussian. For $\beta > \tilde{\beta}_c$, these finite-size fluctuations become non-Gaussian, signalling a gradual crossover toward the broken-LLN regime rather than a sharp transition exactly at $\beta_c$. This mechanism accounts for the discrepancies between theory and simulation observed near the critical inverse temperature in fig. 2.

**Rank Collapse.** The phase diagram shown in fig. 2 exhibits a phase transition in the cosine similarity between tokens after self-attention. For small initialisation $\beta < \beta_c$, self-attention operates in a *'spread attention' phase* where the attention layer effectively outputs the average of all tokens, making the pair-wise cosine similarity between tokens saturate at one, resulting in a rank-one representation matrix. In the absence of skip connections, this behaviour leads to rank collapse, which makes the Transformer untrainable. This behaviour is also reminiscent of the "clustering" property analysed by Geshkovski et al. (2023b); Bruno et al. (2025); Chen et al. (2025).

**Entropy Collapse.** For $\beta > \beta_c$, the self-attention layer preserves diversity, and hence information, among input tokens, suggesting this regime as a viable initialisation. However, the non-vanishing value of the IPR for $\beta > \beta_c$, eq. (12), reveals that self-attention is in a *localised phase* in this regime: it only attends to a few tokens which are determined by initialisation, rather than learnt, leading to the training instabilities observed by Zhai et al. (2023). In other words, for $\beta > \beta_c$ self-attention suffers from entropy collapse, which cannot be avoided using skip connections. This behaviour is a central novelty of our analysis: Noci et al. (2022) assumes the model operates entirely in the rank-collapse phase, thereby overlooking this distinct failure mode, while the "annealed" approximation of Cowsik et al. (2024) does not capture the large-deviation behaviour underlying entropy collapse.

**Take-Home Message.** The main implication of result 1 is that the only viable initialisation for self-attention lies in the small-variance regime $\beta < \beta_c$, supplemented by skip connections to maintain information flow. To illustrate this, we train a one-layer Transformer on a masked language modelling task at varying $\beta$. As shown in fig. 3, the two phases – spread vs. localised – exhibit distinct training behaviours. In the low-variance regime, skip connections help mitigate rank collapse,

enabling successful training. In contrast, in the high-variance regime, the attention vectors' entropy collapses, and the model fails to train effectively, as escaping the pathological initialisation and learning meaningful patterns would require prohibitively long training. Overall, this result unifies two previously observed phenomena, namely *rank* and *entropy collapse*, within a single theoretical framework: the onset of either phenomenon is separated by a sharp phase transition, governed by the variance of the query and key weight initialisation, as parametrised by $\beta$.

**Sequence Geometry in Depth.** We assumed i.i.d. token embeddings, which implies that before any processing the sequence forms a simple random geometry: by concentration of measure, the tokens lie near the vertices of a high-dimensional simplex, with $q_{tt} \simeq 1$ and $q_{ts} \simeq 0$. As the sequence is propagated through a Transformer at initialisation, this *simplex structure* is preserved in the long-sequence limit (i.e. $q_{tt} \simeq q$ and $q_{ts} \simeq p$), as shown in section B.2.3, although the values of $q$ and $p$ themselves evolve with depth. Thus, result 1 provides a way to track how the geometry changes across layers: given the geometric pair $(q, p)$ at layer $l$, we plug it into the update map to obtain the new pair—and hence the updated geometry—at layer $l + 1$.

Before introducing the remaining Transformer components—residual connections, MLPs, and layer normalisation—we analyse the backward pass through the self-attention layer.

## 3.3 THE BACKWARD PASS

To complete our theory of signal propagation, we derive the following result on the norm of the gradients of query and key weights at initialisation (see section B.4 for the derivation):

**Result 2.** *(Query/Key Gradient Analysis) In the limit $T \to \infty$, under the same hypothesis of result 1, the expected squared Frobenius norm of the query gradient, and analogously for the key, is given by*

$$\frac{T}{d^2 \sqrt{\log T}} \mathbb{E} \left\| \frac{\partial \mathcal{L}}{\partial W_Q} \right\|_F^2 = C \beta \sigma_v^2 \, q(q - p) \left[ (q - p)(Y^{(2)} - 2Y^{(3)}) + p(Y^{(2)})^2 \right], \qquad (13)$$

*where $C$ is a constant independent of $T$ and $d$. The same result holds for $W_K$.*

**Vanishing Gradients.** Result 2 shows that gradients can vanish under two conditions. First, we see that if $q = p$, i.e. if attention is uniform and maps all input tokens into the same output token, gradients vanish. In this case, we recover the well-known result of Noci et al. (2022) that showed that if the inputs to the self-attention layer are already collapsed, and hence all the tokens are the same, gradients vanish. Result 2 goes beyond their result to show that even if input tokens are diverse, i.e. $p \neq q$, gradients vanish if $\beta < \beta_c$ because in that regime, the inverse participation ratios $Y^{(2)}$ and $Y^{(3)}$ tend to zero as the sequence length $T \to \infty$. For long but finite sequences, finite-size effects reduce the threshold down to $\beta < \tilde{\beta}_c = \beta_c/2$ (see section 3.2). We numerically verify result 2 in fig. 3c, where we show that (1) the curves of gradient norm versus initialisation strength collapse with the scaling suggested by result 2, and that (2) gradients do indeed tend to zero for $\beta < \tilde{\beta}_c$.

Our analysis of the backward pass raises a paradox: initialise with $\beta < \beta_c$, and gradients vanish; initialise with $\beta > \beta_c$, and self-attention is stuck with entropy collapse. We discussed in the previous section that entropy collapse can only be avoided by initialising with small weights $\beta < \beta_c$ and adding skip connections. So where do the gradients come from? Result 2 relies on the simplex structure of the embeddings. While this structure holds exactly at initialisation, the embeddings receive a non-zero gradient through the skip connections already in the first backward pass, independently of the value of $\beta$. These updates break the simplex structure on which our derivations depend, and thus also invalidate the vanishing-gradient result, which applies only at initialisation.

## 4 FULL TRANSFORMER BLOCK ANALYSIS

We now describe how to track signal propagation through skip connections, LayerNorm, and the MLP, and then show how these components combine into simple iterative algorithms. Such algorithms allow practitioners to predict the evolution of the average cosine similarity across Transformer layers and to generate *trainability diagrams* that reveal viable hyper-parameter regimes for a given Transformer architecture. An example for BERT-style post-norm Transformers is given in algorithm 1; additional variants, including pre-norm, are reported in section B.5.2.

**Self-attention layer.** The order parameters characterizing sequence geometry evolve in the self-attention layer as derived in section B.2.3:

$$q_{\mathcal{S}} = \sigma_v^2 \left( p + (q - p)\, Y^{(2)}(\beta, q, p) \right), \qquad p_{\mathcal{S}} = \sigma_v^2 p, \tag{14}$$

where we recall that $\mathbb{E}[W_V^\top W_V] = \sigma_V^2 \mathbb{I}_d$. Taking the ratio of these expressions recovers the cosine-similarity update of result 1.

**Adding residual connections.** Because $W_V$ is independent of the residual stream, adding a residual connection of strength $\alpha_{\mathrm{SA}}$ contributes additively (see section B.5.1):

$$q \leftarrow q_{\mathcal{S}} + q\,\alpha_{\mathrm{SA}}^2, \qquad p \leftarrow p_{\mathcal{S}} + p\,\alpha_{\mathrm{SA}}^2. \tag{15}$$

An analogous relation holds for the MLP residual path.

**Adding MLPs.** How the signal propagates through the MLP layers follows from the results of Poole et al. (2016); Schoenholz et al. (2016). For a two-layer ReLU MLP, the evolution of squared norms $q^{(l)}$ and pairwise inner products $p^{(l)}$ is:

$$q^{(l)} = \sigma_w^2 \int \mathcal{D}z\, \phi\!\left(\sqrt{q^{(l-1)}}\, z\right)^2 + \sigma_b^2, \qquad l = 2, \dots, L,$$

$$p^{(l)} = \sigma_w^2 \int \mathcal{D}\mathbf{z}\, \phi\!\left(\sqrt{q^{(l-1)}}\, z_1\right) \phi\!\left(\sqrt{q^{(l-1)}}\, z_2\right) + \sigma_b^2,$$

where $\phi$ is the activation function and $\mathbf{z} = (z_1, z_2)^\top$ is a pair of standard Gaussian variables with covariance $\rho^{(l-1)} = p^{(l-1)}/q^{(l-1)}$. The initial conditions after the first linear layer are $q^{(1)} = q\,\sigma_w^2 + \sigma_b^2$ and $p^{(1)} = p\,\sigma_w^2 + \sigma_b^2$. For a two-layer ReLU MLP, the second-layer outputs simplify to:

$$q^{(2)} = \frac{\sigma_w^2}{2} q^{(1)} + \sigma_b^2, \quad p^{(2)} = \frac{\sigma_w^2}{2} q^{(1)} f(\rho^{(1)}) + \sigma_b^2, \tag{16}$$

where $f(\rho)$ captures correlations after the ReLU nonlinearity Cho and Saul (2009):

$$f(\rho) = \frac{1}{\pi}\left(\sqrt{1 - \rho^2} + \rho\,(\pi - \arccos(\rho))\right).$$

After including the MLP residual connection, the updated quantities become:

$$q \leftarrow q^{(2)} + q\,\alpha_{\mathrm{MLP}}^2, \qquad p \leftarrow p^{(2)} + p\,\alpha_{\mathrm{MLP}}^2. \tag{17}$$

Taking their ratio gives the full block-level update of the average cosine similarity.

**LayerNorm.** The remaining component is layer normalisation, which simply enforces

$$p \leftarrow \frac{p}{q}, \qquad q \leftarrow 1.$$

This update should be applied at the position dictated by the specific Transformer variant under consideration.

---

**Algorithm 1** Post-norm Block Update

1: **Inputs:** $\beta, q, p, \alpha_{\mathrm{SA}}, \alpha_{\mathrm{MLP}}, \sigma_w^2, \sigma_b^2, \sigma_v^2$
2: ▷ Attention layer + residual
3: $\beta_c \leftarrow \sqrt{\frac{2}{q(q-p)}}$
4: $Y^{(2)}(\beta) \leftarrow \max(0, 1 - \beta_c/\beta)$
5: $q \leftarrow \sigma_v^2 \left( p + (q-p) \cdot Y^{(2)}(\beta, q, p) \right) + q \cdot \alpha_{\mathrm{SA}^2}$
6: $p \leftarrow p \cdot (\sigma_v^2 + \alpha_{\mathrm{SA}}^2)$
7: ▷ Post-norm LN
8: $p \leftarrow p/q; \quad q \leftarrow 1$
9: ▷ MLP + residual
10: $q_1 \leftarrow \sigma_w^2 q + \sigma_b^2; \quad p_1 \leftarrow \sigma_w^2 p + \sigma_b^2$
11: $q_2 \leftarrow \frac{\sigma_w^2}{2} q_1 + \sigma_b^2; \quad p_2 \leftarrow \frac{\sigma_w^2}{2} f(p_1/q_1) q_1 + \sigma_b^2$
12: $q \leftarrow q_2 + \alpha_{\mathrm{MLP}}^2 q; \quad p \leftarrow p_2 + \alpha_{\mathrm{MLP}}^2 p$
13: ▷ Post-norm LN
14: $p \leftarrow p/q; \quad q \leftarrow 1$
15: **return** $(q, p)$

---

We conclude the paper by outlining several applications of the resulting algorithm.

## 5 APPLICATIONS

To showcase the versatility of our approach, we consider three case studies: signal propagation in a standard BERT architecture; comparing different placements of LayerNorm; and comparing variations of the self-attention mechanism itself.

**Signal propagation in a vanilla Transformer.** We first used algorithm 1 to analyse signal propagation in a BERT-style Transformer (Devlin et al., 2019). Since BERT uses the post-norm convention for LayerNorm, we state the algorithm 1 for the post-norm architecture; see algorithm 2 for the pre-norm

version. The algorithm states the update for the average norm $q$ and average overlap $p$; the average cosine similarity can be read off after each block simply as $\rho = p/q$. Iterating the algorithm for different values of $\beta$ and $\alpha_{\mathrm{SA}}$ finally yields the trainability diagram shown in fig. 1(d).

**Placement of LayerNorm.** Xiong et al. (2020) showed that placing LayerNorm before the self-attention layer and before the MLP greatly stabilises the training of deep Transformers. Comparing signal propagation using our algorithm, we show in fig. 4 that rank collapse, corresponding to an average token similarity of $\langle \rho \rangle = 1$, does indeed occur much later for pre-LN than for post-LN, confirming pre-LN as the more stable choice.

**Avoiding All Collapses: gain-controlled attention.**
A recent line of work has sought to alleviate rank collapse by directly modifying the self-attention layer itself. Noci et al. (2023); Naderi et al. (2024) proposed to enforce the attention layer to be a perturbation of the identity. Noci et al. (2023) derive an SDE description of the limiting distribution of the overlap (or neural covariance) matrix in the proportional limit where both width and depth go to infinity. To obtain this limit, they introduce a width-dependent temperature parameter and remove layer normalisation. However, layer normalisation is crucial for preventing entropy collapse: the critical value $\beta_c(q, p)$ identified in result 1 depends inversely on the squared average norm $q$. Using our framework, we can show that simply removing the mean of the values along the sequence from the output of the standard self-attention layer, also explored in (Naderi et al., 2024) and reminiscent of gain control in neuroscience, can be combined to great effect with either post-LN or pre-LN. In fig. 4, we show the theoretical prediction for the evolution of the average cosine similarity, illustrating how this modification alleviates rank collapse. Our prelim-

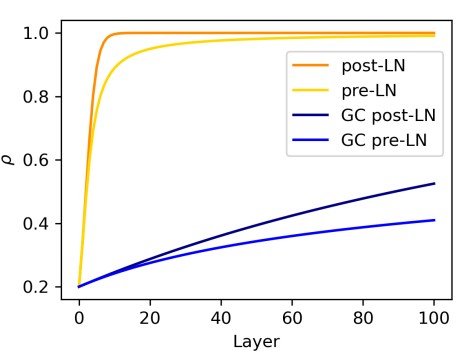

Figure 4: Theoretical prediction of the evolution with depth of the average cosine similarity for the standard Transformer and the Gain-controlled Transformer under both LN strategies. Rank collapse is avoided simply by removing the mean value in the self-attention layer. Here, we set $\alpha_{\mathrm{SA}} = \alpha_{\mathrm{MLP}} = 1$.

inary experiments with a thirty-layer BERT-style Transformer trained on TinyStories show that gain-controlled Transformers succeed in regimes where vanilla attention fails, see fig. 6, encouraging further experiments at scale which are however out of the scope for the present paper. As a final note, in order to propagate the signal infinitely deep, one should simply use gain-controlled attention and initialise the MLPs at the edge of chaos.

**Autoregressive Models.** Our theory relies on the observation that, when starting from embeddings with a simplex structure, non-causal transformers preserve this structure in depth in the limit of infinite sequence length. In this regime, the representation dynamics can be fully described by only two parameters: the average squared norm $q$ and the average overlap $p$. A key ingredient in this result is the infinite-sequence limit itself. For autoregressive models, however, this limit introduces additional subtleties. Early tokens do not satisfy the simplex geometry because they have access to only a small fraction of the sequence context, and in this regime the geometry is more complex and two parameters are no longer sufficient. As one moves to later tokens, the geometry is progressively restored. Consequently, in causal models *rank collapse* does not occur at the level of the full sequence; instead, for sufficiently long sequences, it can emerge in the representations of tokens appearing later in the sequence. Our theory approximates late-position tokens, tracking how their pairwise similarity changes with depth and predicting when their representations collapse (fig. 12).

## CONCLUSIONS

We developed a theory for signal propagation in Transformers that unifies the understanding of the two main failure modes in Transformer training: rank collapse and entropy collapse. Building on these insights, we find new evidence for the viability of a simple architectural modification of self-attention, the gain-controlled self-attention, that avoids both failure modes, which would be interesting to explore at scale in future work.

ACKNOWLEDGEMENTS

We thank Antoine Maillard for helpful discussions. AG gratefully acknowledges funding from Next Generation EU, in the context of the National Recovery and Resilience Plan, Missione 4, Componente 1, Investimento 4.1 "Estensione del numero di dottorati di ricerca e dottorati innovativi per la Pubblica Amministrazione e il patrimonio culturale" (CUP G93C23000620003), and also gratefully acknowledges financial support from Fondazione Zegna. SG gratefully acknowledges funding from the European Research Council (ERC) for the project "beyond2", ID 101166056; from the European Union–NextGenerationEU, in the framework of the PRIN Project SELF-MADE (code 2022E3WYTY – CUP G53D23000780001), and from Next Generation EU, in the context of the National Recovery and Resilience Plan, Investment PE1 – Project FAIR "Future Artificial Intelligence Research" (CUP G53C22000440006).

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

## A  FIGURE DETAILS

All experiments have been conducted using a single NVIDIA A100-PCIE-40GB. Training times vary from approximately one hour (one layer models) to one day (60 layer models).

### A.1  FIGURE 1

**(a, b)**  Attention visualizations obtained using `BertViz` (Vig, 2019) on a single-layer, single-head Transformer at initialization. The attention maps illustrate the effect of varying $\beta$ (directly related to variance of queries/keys): in yellow, a model initialized with $\beta = 0.1$ (low-variance regime, resulting in approximately uniform attention distributions), and in red, a model initialized with $\beta = 1.8$ (high-variance regime, leading to sharp attention).

**(c) Trainability Diagram for a 60-Layer BERT Transformer.**  The diagram is based on the critical value $\beta_c = \sqrt{2}$, which marks the threshold at which entropy collapse occurs in the first self-attention layer under the assumption of i.i.d. and normalised token embeddings. In post-norm architectures (such as BERT), we always have $q = 1$ at the self-attention layer, so $\beta_c = \sqrt{2/(1-\rho)}$. Since $\rho$ increases with depth, the smallest critical value is the one at the first layer (where $\rho \simeq 0$), and we therefore use this value as a single critical threshold for the entire network (i.e. all layers are initialised with the same variance). In other Transformer variants, such as pre-norm, the critical threshold also depends on the average squared norm, so one must determine the appropriate critical value for each layer individually.

The residual-strength threshold is defined as the smallest value of the residual scaling factor $\alpha_{\text{SA}}$ below which rank collapse (loss of representation diversity) occurs across all 60 layers.

**(d)**  Evolution of cosine similarity between token embeddings across layers in a 60-layer Transformer initialized in the low-variance regime (using standard HuggingFace initialization for queries and keys, which corresponds to small $\beta$. Lines denote theoretical predictions, while dots indicate empirical averages over 10 random initializations and 10 input sequences. (Error bars are the standard deviation.) High similarity indicates representational collapse.

**(e)**  Same as (d), but for a 12-layer Transformer initialized in the high-variance regime ($\beta = 1.8$). Again, lines show theoretical predictions and dots indicate empirical means. *Remark:* we intentionally use a shallower model in this regime to ensure that any training failure observed in panel (f) is attributable to entropy collapse rather than rank collapse, as signal propagation across 12 layers is guaranteed.

**(f)**  Pre-training results for BERT-style encoder models using a masked language modeling task (masking probability = 0.15) on the TinyStories dataset. We compare models with 60 layers (blue and yellow: small $\beta \simeq 0.02 < \beta_c$) and 12 layers (red: $\beta = 1.8 > \beta_c$). In particular learning curves in the three phases are: *trainable* ($\beta \simeq 0.02$, $\alpha_{\text{SA}} = 1.5, 2$), *rank collapse* ($\beta \simeq 0.02, \alpha_{\text{SA}} = 1.0$), *entropy collapse* ($\beta = 1.8, \alpha_{\text{SA}} = 1.0, 1.5, 2.0$). Remark: $\beta \simeq 0.02$ corresponds to the standard initialization from Hugginface given the set of hyper-paramters we are using. All models use ReLU activations, 6 attention heads, embedding dimension $d = 600$, and absolute positional embeddings.

Initialization: $\sigma_w^2 = 0.2$, $\sigma_b^2 = 0.0004$, $\sigma_V^2 = \sigma_w^2$; standard HuggingFace initialization for queries/keys; no biases or affine transformations in LayerNorm.

Residual scaling: $\alpha_{\mathrm{SA}}$ as shown in the figure, $\alpha_{\mathrm{MLP}} = 1.0$.

Optimizer:AdamW with learning_rate=1e-4, num_train_epochs=1, batch_size=64, max_grad_norm=1.0, lr_scheduler_type="linear", weight_decay=0.01, and warmup_ratio=0.05.

*Note:* $\beta_c = \beta_c(\rho = 0) = \sqrt{2}$ is the critical threshold for the first layer of self-attention, which at initialization takes inputs that are approximately orthogonal.

## A.2 FIGURE 2

**(Left)** Theoretical prediction.

**(Right)** Cosine similarity averaged over all pairs of tokens when passing a sequence of length $T = 1024$ with embedding dimension $d = 600$ through a single self-attention head.

## A.3 FIGURE 3

Pre-training 1-layer Transformer with 1 head using masked language modeling (masking probability = 0.15) on TinyStories. Embedding dim $d = 768$, standard residual strengths, ReLU activation, absolute position embedding. Custom init of queries/keys with $\beta$s as in figure; standard init for other weights, no biases for queries/keys, no affine in LayerNorm. Optimizer: AdamW, learning_rate=5e-4, num_train_epochs=1, batch_size=64, max_grad_norm=1.0, lr_scheduler_type="linear", weight_decay=0.01, warmup_ratio=0.05.

## B THEORY APPENDIX

### B.1 ATTENTION SCORES ARE CORRELATED GAUSSIAN VARIABLES.

Consider the attention scores defined in eq. (2). By the Central Limit Theorem, they converge in distribution to Gaussian random variables with zero mean and variance $\sigma_a^2$, where $d$ is the embedding dimension (or head dimension in the multi-head case), and $\sigma_a^2$ is determined by the initialisation of $W_Q$ and $W_K$ as described in section 2. Although the scores $a_{tt'}$ are individually Gaussian in the infinite width limit ($d \to \infty$), they are not independent; in fact, they are correlated. To quantify these correlations, we compute:

$$\mathrm{Cov}(a_{ts}, a_{\tau\sigma}) = \frac{1}{d} \sum_{i,j,k,l,m,n=1}^{d} X_{ti} X_{sk} X_{\tau l} X_{\sigma n} \, \mathbb{E}[(W_Q)_{ji}(W_Q)_{ml}] \, \mathbb{E}[(W_K)_{jk}(W_K)_{mn}].$$

Since the query and key weights are independently initialised with variances $\sigma_Q^2 = \sigma_K^2 = \sigma_a/d$, this simplifies to:

$$\mathbb{E}\left[a_{ts} a_{\tau\sigma}\right] = \sigma_Q^2 \sigma_K^2 \left(X_t \cdot X_\tau\right)(X_s \cdot X_\sigma) = \sigma_a^2 q_{t\tau} q_{s\sigma}. \tag{18}$$

### B.2 DERIVATION OF RESULT 1

#### B.2.1 COMPUTATION OF $Y^{(2)}(\beta)$

Consider the partecipation ratio of the $t$-th row of a self-attention matrix average over the initialisation $W_Q$, $W_K$.

$$\mathbb{E} Y_t^{(2)} = \mathbb{E} \frac{\sum_s e^{2a_{ts}}}{\left(\sum_s e^{a_{ts}}\right)^2}$$

We also define

$$\Phi_t(\beta, h) = \mathbb{E} \log Z_t(\beta, h) \tag{19}$$

where

$$Z_t(\beta, h) = \sum_{s=1}^{T} e^{h a_{ts}}, \quad h \in \mathbb{R}$$

and we recall that $\beta$ enters in the definition of the covariances between the $a$'s, as they are all proportional to $\sigma_a^2 = \beta^2 \log T$.

We want to exploit Stein's lemma, which yields:

$$\partial_h \Phi_t(\beta, h) = \mathbb{E}\left[ \frac{\sum_s a_{ts} e^{ha_{ts}}}{\sum_u e^{ha_{tu}}} \right]$$

$$= \sum_{s,s'=1}^{T} \mathbb{E}[a_{ts} a_{ts'}] \, \mathbb{E}\left[ \partial_{a_{ts'}} \left( \frac{e^{ha_{ts}}}{\sum_u e^{ha_{tu}}} \right) \right] \tag{20}$$

with the correlation between attention scores given by eq. (18). We assume $q_{tt} \simeq q$ for all $t$, due to concentration of measure. For the pairwise overlaps, we proceed as follows: at the first layer, all tokens are approximately orthogonal, making it safe to assume $q_{ts} \simeq 0 \ \forall t \neq s \in [T]$. One then derives the update for this layer and observes that, in the limit of infinite sequence length (as we will briefly show), the update becomes independent of the indices to leading order. By repeating this argument across layers, it is therefore justified to treat the $q_{ts}$ as having a common mean $p$ together with sub-leading Gaussian fluctuations, which we neglect since our analysis focuses on the average overlap. Applying this argument, we get:

$$\partial_h \Phi_t(\beta, h) \simeq h\sigma_a^2 q \sum_s \mathbb{E}\left( q \frac{e^{ha_{ts}}}{\sum_u e^{ha_{tu}}} - q \frac{e^{2ha_{ts}}}{(\sum_u e^{ha_{tu}})^2} - p \sum_{s' \neq s} \frac{e^{ha_{ts}} e^{ha_{ts'}}}{(\sum_u e^{ha_{tu}})^2} \right)$$

$$= h\sigma_a^2 q \, (q - p) \left( 1 - \mathbb{E}Y_t^{(2)} \right)$$

Finally, this leads to:

$$\lim_{T \to \infty} \mathbb{E}Y_t^{(2)} = 1 - \frac{1}{\sigma_a^2 q \, (q - p)} \lim_{h \to 1} \lim_{T \to \infty} \partial_h \Phi_t(\beta, h) \tag{21}$$

To proceed, we are left with computing the expectation $\Phi_t(\beta, h) = \mathbb{E}\left[ \log \sum_s e^{ha_{ts}} \right]$.

The remaining computation amounts to evaluating a variant of the Random Energy Model (REM), but with correlated energy levels due to the structure of the $a_{ts}$ variables. This problem can be tackled using the Replica method or, alternatively, via a micro-canonical argument, as discussed in Mézard and Montanari (2009). Here, we proceed with the Replica method.

We compute the replicated partition function as:

$$\mathbb{E}_a Z_t^n(\beta, h) = \mathbb{E}_a\left[ \left( \sum_s e^{ha_{ts}} \right)^n \right] = \sum_{s_1,\ldots,s_n=1}^{T} \mathbb{E}_a\left[ \exp\left( h \sum_{a=1}^{n} a_{ts_a} \right) \right]$$

$$= \sum_{s_1,\ldots,s_n=1}^{T} \exp\left( \frac{h^2 \sigma_a^2}{2} \sum_{a,b=1}^{n} \mathbb{E}[a_{ts_a} a_{ts_b}] \right)$$

$$= \sum_{s_1,\ldots,s_n=1}^{T} \exp\left( \frac{h^2 \sigma_a^2}{2} \left( q \sum_{a,b} \mathbb{I}(s_a = s_b) + qp \sum_{s \neq s'} \sum_{a,b} \mathbb{I}(s = s_a)\mathbb{I}(s' = s_b) \right) \right) \tag{22}$$

We now introduce the empirical overlap matrix:

$$Q_{ab} = \mathbb{I}(s_a = s_b)$$

and perform a change of variables from replica indices to overlap structures $Q$, giving:

$$\mathbb{E}_a Z_t^n(\beta, h) = \sum_Q \sum_{\{s_a\}} \prod_{a,b} \delta(Q_{ab}, \mathbb{I}(s_a = s_b)) \exp\left( \frac{h^2 \sigma_a^2}{2} q \, (q - p) \sum_{a,b} Q_{ab} + O(n^2) \right)$$

$$= \sum_Q S(Q) \exp\left( \frac{h^2 \sigma_a^2}{2} q \, (q - p) \sum_{a,b} Q_{ab} + O(n^2) \right) \tag{23}$$

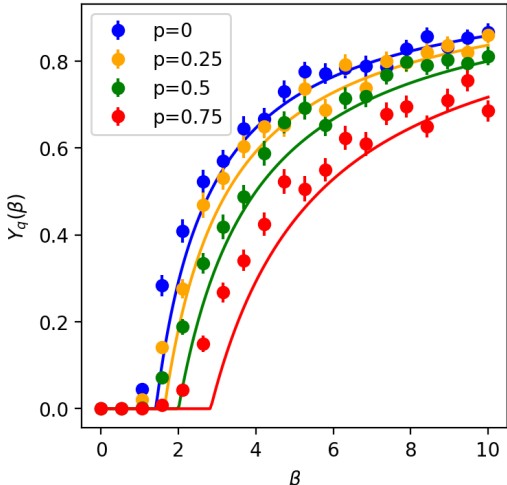

Figure 5: Theory and experiments ($T = 10^5$) comparison of the computation of $Y^{(2)}(\beta)$, finite size effects are visible around the phase transition.

Now we take the 1-RSB ansatz for $Q$: the $n$ replicas are divided into $\frac{n}{x}$ groups of $x$ elements which are in the same energy configurations.
Moreover, we need to consider exponentially long sequences, i.e. we take $T = e^N$ and control $N$. This implies:

$$S(Q) \simeq e^{N\frac{n}{x}} \qquad \sum_{ab}^{n} Q_{ab} = nx$$

So exploiting the replica trick and recalling the definition of $\sigma_a^2$, we get:

$$\Phi_t(\beta, h)/N = \max_{x<1} \frac{\beta^2 h^2}{2} q\,(q-p)\,x + \frac{1}{x} \tag{24}$$

which leads to the existence of a critical temperature $\beta_c(h, q, p) = \frac{1}{h}\sqrt{\frac{2}{q(q-p)}}$ where a condensation phase transition takes place. In particular we have:

$$\Phi_t(\beta, h)/N = \begin{cases} 1 + \frac{\beta^2 h^2}{2} q\,(q-p) & \beta < \beta_c(h, q, p) \\ \beta h \sqrt{2q\,(q-p)} & \beta > \beta_c(h, q, p) \end{cases}$$

Due to the fact that we took the expectation over all the $a$'s there is no dependence on $v$, rather only on the similarity matrix, and so we can drop the subscript from $\Phi_t$. Finally we can put all together, and we get:

$$\lim_{T\to\infty} \mathbb{E} Y_t(\beta) = 1 - \frac{1}{\sigma_a^2 q\,(q-p)} \lim_{h\to 1} \lim_{T\to\infty} \partial_h \Phi(\beta, h) := Y^{(2)}(\beta) = \begin{cases} 0 & \beta < \beta_c(q, p) \\ 1 - \frac{\beta_c(q,p)}{\beta} & \beta > \beta_c(q, p) \end{cases}$$

where $\beta_c(q, p) = \beta_c(h = 1, q, p) = \sqrt{\frac{2}{q(q-p)}}$. We plot in fig. 5 the result of the computation and some numerical simulations.

### B.2.2 COMPUTATION OF $Y_p(\beta)$

We need to compute:

$$Y_p(\beta) = \mathbb{E} \frac{\sum_s e^{a_{vs}+a_{us}}}{\sum_s e^{a_{vs}} \sum_{s'} e^{a_{us'}}}$$

Consider the partition function with auxiliary fields $\mathbf{h} = (h_{ss'})_{s,s'=1}^T$:

$$Z_{v,u}(\beta, \mathbf{h}) = \sum_{ss'} e^{h_{ss'}(a_{vs}+a_{us'})}$$

Let's observe that:

$$\sum_\sigma \partial_{h_{\sigma\sigma}} \mathbb{E} \log Z_{v,u}(\beta, \mathbf{h}) = \mathbb{E}\left[ \frac{\sum_\sigma (a_{v\sigma} + a_{u\sigma}) e^{h_{\sigma\sigma}(a_{v\sigma} + a_{u\sigma})}}{\sum_{s,s'} e^{h_{ss'}(a_{vs} + a_{us'})}} \right]$$

$$= \mathbb{E}\left[ \frac{\sum_\sigma (a_{v\sigma} + a_{u\sigma}) e^{h_{\sigma\sigma}(a_{v\sigma} + a_{u\sigma})}}{\sum_\sigma e^{h_{\sigma\sigma}(a_{v\sigma} + a_{u\sigma})}} \cdot \frac{\sum_\sigma e^{h_{\sigma\sigma}(a_{v\sigma} + a_{u\sigma})}}{\sum_{s,s'} e^{h_{ss'}(a_{vs} + a_{us'})}} \right] \quad (25)$$

Assuming the first term is self-averaging, we approximate:

$$\sum_\sigma \partial_{h_{\sigma\sigma}} \mathbb{E} \log Z_{v,u}(\beta, \mathbf{h}) \simeq \mathbb{E}\left[ \frac{\sum_\sigma (a_{v\sigma} + a_{u\sigma}) e^{h_{\sigma\sigma}(a_{v\sigma} + a_{u\sigma})}}{\sum_\sigma e^{h_{\sigma\sigma}(a_{v\sigma} + a_{u\sigma})}} \right] \cdot \mathbb{E}\left[ \frac{\sum_\sigma e^{h_{\sigma\sigma}(a_{v\sigma} + a_{u\sigma})}}{\sum_{s,s'} e^{h_{ss'}(a_{vs} + a_{us'})}} \right]$$

Under this approximation:

$$Y_p(\beta) = \frac{\lim_{\mathbf{h}\to 1} \sum_\sigma \partial_{h_{\sigma\sigma}} \mathbb{E} \log \sum_{s,s'} e^{h_{ss'}(a_{vs} + a_{us'})}}{\lim_{h\to 1} \partial_h \mathbb{E} \log \sum_s e^{h(a_{vs} + a_{us})}} \quad (26)$$

The computation of the free entropy in the denominator is straightforward. It closely resembles the derivation for the free entropy appearing in the calculation of $Y_q(\beta)$.

One gets:

$$\frac{1}{N} \mathbb{E} \log \sum_s e^{h(a_{vs} + a_{us})} = \begin{cases} 1 + \left(1 - \frac{p^2}{q^2}\right)\beta^2 h^2 & \beta < \frac{1}{h\sqrt{1 - \frac{p^2}{q^2}}} \\ 2\beta h \sqrt{1 - \frac{p^2}{q^2}} & \beta > \frac{1}{h\sqrt{1 - \frac{p^2}{q^2}}} \end{cases} \quad (27)$$

which gives

$$\lim_{h\to 1} \partial_h \mathbb{E} \log \sum_s e^{h(a_{vs} + a_{us})} = \begin{cases} 2\beta^2 \left(1 - \frac{p^2}{q^2}\right) & \beta < \frac{1}{\sqrt{1 - \frac{p^2}{q^2}}} \\ 2\beta \sqrt{1 - \frac{p^2}{q^2}} & \beta > \frac{1}{\sqrt{1 - \frac{p^2}{q^2}}} \end{cases} \quad (28)$$

The calculation of the free entropy at the numerator is a bit more involved, but we can observe the following:

$$\lim_{\mathbf{h}\to 1} \sum_\sigma \partial_{h_{\sigma\sigma}} \mathbb{E} \log \sum_{s,s'} e^{h_{ss'}(a_{vs} + a_{us'})} = \frac{\sum_s (a_{vs} + a_{us}) e^{a_{vs} + a_{us}}}{\sum_{s,s'} e^{a_{vs} + a_{us'}}}$$

$$= \frac{\sum_{s,s'} (a_{vs} + a_{us'}) e^{a_{vs} + a_{us'}} - \sum_{\substack{s,s'=1 \\ s\neq s'}} (a_{vs} + a_{us'}) e^{a_{vs} + a_{us'}}}{\sum_{s,s'} e^{a_{vs} + a_{us'}}}$$

We define:

$$\langle a_{vs} \rangle_s = \frac{\sum_s a_{vs} e^{a_{vs}}}{\sum_s e^{a_{vs}}}$$

$$\langle (a_{vs} + a_{us'}) \rangle_{s\neq s'} = \frac{\sum_{\substack{s,s'=1 \\ s\neq s'}} (a_{vs} + a_{us'}) e^{a_{vs} + a_{us'}}}{\sum_{\substack{s,s'=1 \\ s\neq s'}} e^{a_{vs} + a_{us'}}}$$

Thus, we obtain the approximation:

$$\lim_{\mathbf{h}\to\mathbf{1}} \sum_{\sigma} \partial_{h_{\sigma\sigma}} \mathbb{E} \log \sum_{s,s'} e^{h_{ss'}(a_{vs}+a_{us'})} \leq \langle a_{vs}\rangle_s + \langle a_{us}\rangle_s - \langle(a_{vs}+a_{us'})\rangle_{s\neq s'} \simeq 0$$

There is also an intuitive way to see this. Consider the two limiting cases:

- As $\beta \to 0$: The attention weights become uniform, i.e., $A_{vs} \to \frac{1}{T}$. Then,

$$\frac{1}{d}\mathbb{E}_{QKV}\left[\mathcal{S}(X)_v \cdot \mathcal{S}(X)_w\right] = \mathbb{E}_{KQ}\sum_{s,\sigma} A_{vs}A_{w\sigma}S_{s\sigma} \to \frac{1}{T^2}\sum_{s,\sigma}S_{s\sigma} = \frac{p}{q} + \mathcal{O}(T^{-1}),$$

  meaning both $q^{\text{att}}$, $p^{\text{att}} \to \frac{p}{q}$.

- As $\beta \to \infty$: The attention becomes fully peaked:

$$A_{vs} \to \delta_{v,s^*(v)}, \quad \text{with} \quad s^*(v) = \arg\max_s a_{vs}.$$

  In this limit, the dot product becomes:

$$q^{\text{att}} \to S_{s^*s^*} = 1, \quad p^{\text{att}} \to S_{s^*(v),\sigma^*(w)} = \frac{p}{q},$$

  since $s^*(v) \neq \sigma^*(w)$ with probability $\pi = 1 - 1/T$.

Hence, as we vary $\beta$ from 0 to $\infty$, the quantity $\mathrm{SA}(q)$ interpolates between $\frac{p}{q}$ and 1, while $\mathrm{SA}(p)$ remains nearly constant.

### B.2.3 UPDATE MAP OF THE AVERAGE AVERAGE COSINE SIMILARITY.

We begin our analysis by averaging over the value projection matrix $W_V$. Since $W_V$ is independent from $W_Q$, $Q_K$ at initialisation, the scalar product between self-attention outputs then becomes:

$$\frac{1}{d}\mathbb{E}\left[\mathcal{S}(X)_t \cdot \mathcal{S}(X)_{t'}\right] = \frac{1}{d}\mathbb{E}\left[\sum_{s,\sigma} A_{ts}A_{t'\sigma}X_s^\top \, \mathbb{E}[W_V^\top W_V]\, X_\sigma\right].$$

Choosing, without loss of generality, the variance of the value projection weights as $\sigma_V^2 = 1/d$, we obtain $\mathbb{E}_V[W_V^\top W_V] = \mathbb{I}_d$, so the expression simplifies to:

$$\mathbb{E}\left[\sum_{s,\sigma} A_{ts}A_{t'\sigma}q_{s\sigma}\right]. \tag{29}$$

With the limit $T \to \infty$ in mind, we neglect sub-leading fluctuations and focus on the leading terms:

$$q_{s\sigma} \simeq p \quad \text{for } s \neq \sigma, \qquad q_{ss} \simeq q\,.$$

This approximation holds due to concentration of measure immediately after the embedding layer. Moreover, the result we derive in the infinite-sequence-length limit for the first layer is independent of the specific token indices, meaning that the simplex geometry is preserved. Consequently, the same reasoning applies recursively at every layer, implying that the dynamics are fully captured by the two scalar quantities $q$ and $p$: they are the only relevant geometric degrees of freedom precisely because the simplex structure persists throughout the network.

$$\mathbb{E}\left[q\sum_s A_{ts}A_{t's} + p\sum_{s\neq\sigma} A_{ts}A_{t'\sigma}\right]. \tag{30}$$

Since each row of the attention matrix sums to one, we can simplify this expression further:

$$q\sum_s A_{ts}A_{t's} + p\left(\sum_{s,\sigma} A_{ts}A_{t'\sigma} - \sum_s A_{ts}A_{t's}\right) = (q-p)\sum_s A_{ts}A_{t's} + p.$$

Now, upon averaging and taking the limit $T \to \infty$, two distinct quantities emerge depending on whether $t = t'$ or $t \neq t'$:

$$Y^{(2)}(\beta) = \lim_{T \to \infty} \mathbb{E}\left[\sum_s A_{ts}^2\right], \qquad Y_p(\beta) = \lim_{T \to \infty} \mathbb{E}\left[\sum_s A_{ts} A_{t's}\right] \quad \text{for } t \neq t', \qquad (31)$$

which quantify the self- and cross-overlap of attention distributions. The expression for the first one is derived in section B.2.1 and in section B.2.2 we show that the second term is sub-leading.

Substituting the result, we get the update for the average squared norm:

$$q \overset{\mathcal{S}}{\leftarrow} p + (q - p) Y_q(\beta) = \begin{cases} p & \beta < \beta_c(q, p) \\ p + (q - p)\left(1 - \frac{\beta_c(q,p)}{\beta}\right) & \beta > \beta_c(q, p) \end{cases} \qquad (32)$$

where $\beta_c(q, p) = \sqrt{\frac{2}{q(q-p)}}$.

On the other hand, the scalar product $p$ is not updated, as $Y_p(\beta)$ is sub-leading. Taking the ratio between the updates yields, at leading order in $d$, the following update for the average cosine similarity:

$$\Phi_{\mathcal{S}}(\rho) = \frac{\rho}{1 + (1 - \rho)Y^{(2)}(\beta)},$$

which gives rise to the phase diagram in fig. 2.

## B.3 FINITE-SIZE EFFECTS

Here we give a non-rigorous argument on the finite size effects that afflict our asymptotic theory. In the low-$\beta$ regime, the attention is spread approximately uniformly over a number $T^* = e^{S(\beta,\rho)}$ of keys, given by an entropic quantity $S(\beta, \rho) = \Phi(\beta, \rho) - \beta \partial_\beta \Phi(\beta, \rho)$ (where the free entropy $\Phi$ was defined in eq. (19)). A derivation of the entropy for the REM, very much related to our problem, is explained in depth by Mézard and Montanari (2009). For $\beta < \beta_c(\rho)$, this turns out to be:

$$S(\beta, \rho) = N\left(1 - \frac{\beta^2}{\beta_c(\rho)}\right) \qquad (33)$$

Since the IPR is the inverse number of the expected number of state that matter:

$$Y^{(2)}(\beta, \rho) \simeq e^{-S(\beta, \rho)} \qquad (34)$$

In the limit $N = \mathcal{O}(\log T)$, this non-rigorous argument suggests that the corrections to eq. (12) scale are $O\left(T^{-1 + \frac{\beta^2}{\beta_c(\rho)^2}}\right)$. As long as $\beta < \tilde{\beta}_c = \beta_c/2$, these fluctuations can be neglected since they remain Gaussian. For $\beta_c/2 < \beta < \beta_c$, however, the fluctuations around the asymptotic solution become non-Gaussian, as the central limit theorem breaks down at $\beta_c/2$.

## B.4 DERIVATION OF RESULT 2

Consider square matrices $A, a, Q \in \mathbb{R}^{T \times T}$, where $A = \text{softmax}(a)$ is the standard attention matrix computed from logits $a$ and $Q = (q_{ts})_{(ts)}$ is the overlap matrix. Let $I_T$ denote the $T \times T$ identity matrix. Also, consider matrices $X \in \mathbb{R}^{T \times d}$, $W_V, W_Q, W_K \in \mathbb{R}^{d \times d}$.

Define the attention operation

$$\mathcal{S}(X) = A X W_V. \qquad (35)$$

We are interested in computing the squared Frobenius norm of the gradient of $\mathcal{S}(X)$ with respect to the query matrix $W_Q$:

$$\left\|\frac{\partial \mathcal{S}(X)}{\partial W_Q}\right\|_F^2 = \text{tr}\left(\frac{\partial \mathcal{S}(X)}{\partial W_Q}\left(\frac{\partial \mathcal{S}(X)}{\partial W_Q}\right)^\top\right). \qquad (36)$$

The first part of the derivation parallels the proof of a related result in Noci et al. (2022). However, unlike their approach, we do not assume uniform attention. Instead, we retain the attention explicitly, which allows the previously derived participation ratio to naturally emerge.

Chain rule decomposition:

$$\frac{\partial \mathcal{S}(X)}{\partial W_Q} = \left(I_T \otimes W_V^\top X^\top\right) \frac{\partial A}{\partial a} \left(\frac{1}{\sqrt{d}} X \otimes X W_K\right).$$

Consequently, the squared Frobenius norm equals the trace of

$$\left(I_T \otimes W_V^\top X^\top\right) \frac{\partial A}{\partial a} \left(\frac{1}{\sqrt{d}} X \otimes X W_K\right)$$

$$\times \left(\frac{1}{\sqrt{d}} X^\top \otimes W_K^\top X^\top\right) \left(\frac{\partial A}{\partial a}\right)^\top (I_T \otimes X W_V).$$

Simplification of the middle terms:

$$\left(\frac{1}{\sqrt{d}} I_T \otimes X W_K\right) \left(\frac{1}{\sqrt{d}} I_T \otimes W_K^\top X^\top\right)$$

$$= \frac{1}{d} I_T \otimes \left(X W_K W_K^\top X^\top\right).$$

Assuming $W_K W_K^\top$ concentrates as

$$W_K W_K^\top \approx d\sigma_K^2 I_d,$$

we obtain

$$\frac{1}{d} X X^\top \otimes \left(X(d\sigma_K^2 I_d) X^\top\right) = \sigma_K^2 \left(X X^\top \otimes X X^\top\right).$$

Taking the trace and using its cyclic property, we define

$$G = \left(\frac{\partial A}{\partial a}\right) \left(X X^\top \otimes X X^\top\right) \left(\frac{\partial A}{\partial a}\right)^\top,$$

and write

$$\left\|\frac{\partial \mathcal{S}(X)}{\partial W_Q}\right\|_F^2 = \sigma_K^2 \operatorname{tr}\left(G \left(I_T \otimes X W_V\right) \left(I_T \otimes W_V^\top X^\top\right)\right).$$

Assuming $W_V W_V^\top$ concentrates as

$$W_V W_V^\top \approx d\sigma_V^2 I_d,$$

we simplify further as

$$\left\|\frac{\partial \mathcal{S}(X)}{\partial W_Q}\right\|_F^2 = d\sigma_K^2 \sigma_V^2 \operatorname{tr}\left(G \left(I_T \otimes X X^\top\right)\right).$$

Recall the definition of the overlap matrix

$$Q = \frac{1}{d} X X^\top \in \mathbb{R}^{T \times T},$$

we get the compact expression

$$\left\|\frac{\partial \mathcal{S}(X)}{\partial W_Q}\right\|_F^2 = d^4 \sigma_K^2 \sigma_V^2 \operatorname{tr}\left(G(I_T \otimes Q)\right),$$

where

$$G = \left(\frac{\partial A}{\partial a}\right)(Q \otimes Q)\left(\frac{\partial A}{\partial a}\right)^{\top}.$$

This expression reveals how the gradient norm depends on the structure of $Q$, the Jacobian of the attention, and the variance parameters associated with the key and value projections.

Let's compute the trace term:

$$\text{tr}\left(\frac{\partial A}{\partial a}(Q \otimes Q)\left(\frac{\partial A}{\partial a}\right)^{\top}(I_T \otimes Q)\right)$$

where $\frac{\partial A}{\partial a}$ is the Jacobian matrix of size $T^2 \times T^2$. Let's write the Jacobian in components. Using the fact that $A = \text{softmax}(a)$, so the Jacobiam components are:

$$D_{(ij),(kl)} := \frac{\partial A_{ij}}{\partial a_{kl}} = \delta_{ik}\delta_{jl}A_{ij} - \delta_{ik}A_{ij}A_{il}. \tag{37}$$

Now, the trace can be written as

$$\text{tr} = \sum_{i,j,r,s=1}^{T}\left[\frac{\partial A}{\partial a}(Q \otimes Q)\left(\frac{\partial A}{\partial a}\right)^{\top}(I_T \otimes Q)\right]_{(ij),(tu)}\delta_{it}\delta_{ju}.$$

Expanding indices leads to

$$\text{tr} = \sum_{i,j,k,l,m,n,r,s,t,u=1}^{T} D_{(ij),(kl)}q_{km}q_{ln}D_{(rs),(mn)}\delta_{rt}q_{su}\delta_{it}\delta_{ju}$$

Simplifying the deltas:

$$\text{tr} = \sum_{i,j,k,l,m,n,s=1}^{T} D_{(ij),(kl)}q_{km}q_{ln}D_{(is),(mn)}q_{sj}$$

Now we can substitute the Jacobian components given by eq. (37).

Assume Einstein's notation.

$$q_{i,i}\left[A_{ij}A_{in}q_{jn}^2 - A_{ij}A_{is}A_{in}q_{jn}q_{sj} - A_{ij}A_{il}A_{in}q_{ln}q_{nj} + A_{ij}A_{il}A_{is}A_{in}q_{ln}q_{sj}\right]$$

To leading order in $d$ we can substitute $Q$ with its expectation value,

$$q_{ts} \simeq p + (q-p)\delta_{ts}.$$

Expanding each contribution and taking the limit:

(1) $A_{ij}A_{in}q_{jn}^2 \longrightarrow p^2 + \left(2p(q-p) + (q-p)^2\right)Y^{(2)},$

(2) $A_{ij}A_{is}A_{in}q_{jn}q_{sj} \longrightarrow p^2 + 2p(q-p)Y^{(2)} + (q-p)^2Y^{(3)},$

(3) $A_{ij}A_{il}A_{in}q_{ln}q_{nj} \longrightarrow p^2 + 2p(q-p)Y^{(2)} + (q-p)^2Y^{(3)},$

(4) $A_{ij}A_{il}A_{is}A_{in}q_{ln}q_{sj} \longrightarrow p^2 + 2p(q-p)Y^{(2)} + p(q-p)\left(Y^{(2)}\right)^2.$

putting all together:

$$tr = q(q-p)\Big[(q-p)\big(Y^{(2)} - 2Y^{(3)}\big) + p\big(Y^{(2)}\big)^2\Big].$$

Finally:

$$\mathbb{E}\left\|\frac{\partial \mathcal{S}(X)}{\partial W_Q}\right\|_F^2 = d^4\sigma_K^2\sigma_V^2\, q\sum_{ij} A_{ij}^2(q-p)^2 + q\sum_{ij} A_{ij}^3(q-p)^2 + q(q-p)^2\sum_{ijn} A_{ij}^2 A_{in}^2 \quad (38)$$

Recall that we took $\sigma_K^2 = \beta\sqrt{\log(T)}/d$ and $\sigma_V^2 = \sigma_v^2/d$, so:

$$\frac{1}{d^2}\mathbb{E}\left\|\frac{\partial \mathcal{S}(X)}{\partial W_Q}\right\|_F^2 = \beta\sqrt{\log(T)}\sigma_v^2\, q(q-p)\Big[(q-p)\big(Y^{(2)} - 2Y^{(3)}\big) + p\big(Y^{(2)}\big)^2\Big]. \quad (39)$$

Now if consider instead the gradient of the loss:

$$\frac{1}{d^2}\mathbb{E}\left\|\frac{\partial \mathcal{L}}{\partial W_Q}\right\|_F^2 \leq \mathcal{B}(X)\beta\sqrt{\log(T)}\sigma_v^2\, q(q-p)\Big[(q-p)\big(Y^{(2)} - 2Y^{(3)}\big) + p\big(Y^{(2)}\big)^2\Big]. \quad (40)$$

where $\mathcal{B}(X)$ is a bounded a quantity of $X$ (see the proof of theorem 3.2 Noci et al. (2022)).

We assume that this quantity scales like $O_T(T^{-1})$ and check it numerically in fig. 3(c). So putting all together:

$$\frac{T}{d^2\sqrt{\log T}}\mathbb{E}\left\|\frac{\partial \mathcal{L}}{\partial W_Q}\right\|_F^2 \propto \beta\sigma_v^2\, q(q-p)\Big[(q-p)\big(Y^{(2)} - 2Y^{(3)}\big) + p\big(Y^{(2)}\big)^2\Big]. \quad (41)$$

Since the $Y^{(r)} \to 0$ in the small $\beta$ regime (Mézard and Montanari, 2009), let's check our predictions for the case where $q = 1$, $p \simeq 0$. In this case $\beta_c = \sqrt{2}$, but in light of the discussion on finite size effects in section 3.2, we actually expect our prediction to be sharp up to $\beta_c/2 \approx 0.7$ and then a crossover between $\beta_c/2$ and $\beta_c$ to the other solution. This behaviour is correctly predicted, see fig. 3(c).

### B.5   Signal Propagation in the full transformer block

#### B.5.1   Action of residual connections

Recall the action of the skip connection in self-attention:

$$\text{RES}_{\text{SA}}(X) = \mathcal{S}(X) + \alpha_{\text{SA}}X = AXW_V + \alpha_{\text{SA}}X.$$

Consider the quantity

$$\mathbb{E}_{Q,K,V}\big[\text{RES}_{\text{SA}}(X)_t \cdot \text{RES}_{\text{SA}}(X)_s\big].$$

The expectation over the value matrix vanishes in the mixed terms, leading to

$$\mathbb{E}_{Q,K,V}\big[\mathcal{S}(X)_t \cdot \mathcal{S}(X)_s\big] + \alpha_{\text{SA}}^2\, X_t \cdot X_s.$$

Overall, considering

$$\rho = \frac{\mathbb{E}\langle q_{ts}\rangle}{\mathbb{E}\langle q_{tt}\rangle} = \frac{\sigma_v^2 p + \alpha_{\text{SA}}^2 p}{\sigma_v^2\big(p + (q-p)Y^{(2)}(\beta)\big) + \alpha_{\text{SA}}^2 q},$$

where we used $\mathbb{E}[W_V^\top W_V] = \sigma_v^2 I_d$ and the updates for $p$ and $q$ described in section B.2.3.

#### B.5.2   Full Block cosine similarity update algorithms

---

**Algorithm 2** Pre-norm Block Update

---

1: **Inputs:** $\beta, q, p, \alpha_{\text{SA}}, \alpha_{\text{MLP}}, \sigma_w^2, \sigma_b^2, \sigma_v^2$
2: $\triangleright$ Pre-norm LN before attention
3: $p_{\text{LN}} \leftarrow p/q; \quad q_{\text{LN}} \leftarrow 1$
4: $\triangleright$ Attention layer (normed input) + residual
5: $\beta_c \leftarrow \sqrt{\frac{2}{1(1 - p_{\text{LN}})}}$
6: $Y^{(2)}(\beta) \leftarrow \max(0, 1 - \beta_c/\beta)$
7: $q \leftarrow \sigma_v^2 \left( p_{\text{LN}} + (q_{\text{LN}} - p_{\text{LN}}) \cdot Y^{(2)}(\beta) \right) + q \cdot \alpha_{\text{SA}}^2$
8: $p \leftarrow \sigma_v^2 p_{\text{LN}} + \alpha_{\text{SA}}^2 p$
9: $\triangleright$ Pre-norm LN before MLP
10: $p_{\text{LN}} \leftarrow p/q; \quad q_{\text{LN}} \leftarrow 1$
11: $\triangleright$ MLP layer (normed input) + residual
12: $q_1 \leftarrow \sigma_w^2 q_{\text{LN}} + \sigma_b^2; \quad p_1 \leftarrow \sigma_w^2 p_{\text{LN}} + \sigma_b^2$
13: $q_2 \leftarrow \frac{\sigma_w^2}{2} q_1 + \sigma_b^2; \quad p_2 \leftarrow \frac{\sigma_w^2}{2} f(p_1/q_1) q_1 + \sigma_b^2$
14: $q \leftarrow q_2 + \alpha_{\text{MLP}}^2 q; \quad p \leftarrow p_2 + \alpha_{\text{MLP}}^2 p$
15: **return** $(q, p)$

---

**Algorithm 3** Gain-controlled Transformer Block Update with Post-norm

---

1: **Inputs:** $\beta, q, p, \alpha_{\text{SA}}, \alpha_{\text{MLP}}, \sigma_w^2, \sigma_b^2, \sigma_v^2$
2: $\triangleright$ Attention layer + residual (centered-value update)
3: $\beta_c \leftarrow \sqrt{\frac{2}{q(q-p)}}$
4: $Y^{(2)}(\beta) \leftarrow \max(0, 1 - \beta_c/\beta)$
5: $q \leftarrow \sigma_v^2 \left( (q - p) Y^{(2)}(\beta) \right) + \alpha_{\text{SA}}^2 q$
6: $p \leftarrow \alpha_{\text{SA}}^2 p$
7: $\triangleright$ Post-norm LN
8: $p \leftarrow p/q; \quad q \leftarrow 1$
9: $\triangleright$ MLP + residual
10: $q_1 \leftarrow \sigma_w^2 q + \sigma_b^2; \quad p_1 \leftarrow \sigma_w^2 p + \sigma_b^2$
11: $q_2 \leftarrow \frac{\sigma_w^2}{2} q_1 + \sigma_b^2; \quad p_2 \leftarrow \frac{\sigma_w^2}{2} f(p_1/q_1) q_1 + \sigma_b^2$
12: $q \leftarrow q_2 + \alpha_{\text{MLP}}^2 q; \quad p \leftarrow p_2 + \alpha_{\text{MLP}}^2 p$
13: $\triangleright$ Post-norm LN
14: $p \leftarrow p/q; \quad q \leftarrow 1$
15: **return** $(q, p)$

---

## C  SUPPLEMENTARY FIGURES

### C.1  TRAINING GAIN-CONTROLLED TRANSFORMERS ON TINYSTORIES.

We train gain-controlled transformers with post-LN with skip connections strength $\alpha_{\text{SA}} = \alpha_{\text{MLP}} = 1$ of one and twenty layers. The latter case would fail for a standard transformer in the same setting due to rank collapse. The training dynamics is reported in fig. 6.

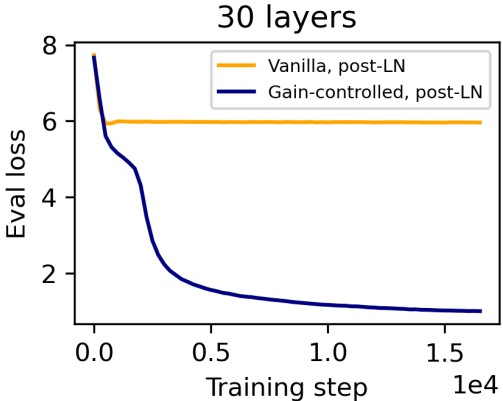

Figure 6: Training 30 layers of vanilla and Gain-controlled Transformer on TinyStories.

Details of training: 30-layer, single-head BERT-style model with embedding size 480 and ReLU activation, using masked language modeling with 15% masking probability, a learning rate of 5e-4, batch size 64, warmup ratio 0.05, weight decay 0.01, for 0.5 epochs.

### C.2  VISUALIZING THE THE QUERY/KEY VARIANCE EFFECT ON THE SPECTRUM OF THE SELF-ATTENTION MATRIX

As shown by Bordenave et al. (2012), the spectral bulk of a random stochastic matrix—such as an attention matrix at initialization—has radius $\mathcal{O}(T^{-1/2})$. This result applies under standard initialization schemes, where the variance of the attention scores remains fixed and independent of sequence length.

In contrast, under our proposed rescaling of the query and key weight matrices, the variance of the attention scores scales as $\sigma_a^2 = \log T$, which depends on the dimension of the attention matrix. This places the system in a different spectral regime and, crucially, avoids the spectral bulk to have $\mathcal{O}(T^{-1/2})$ radius, key driver of rank collapse in conventional transformer models, as visualized in fig. 7.

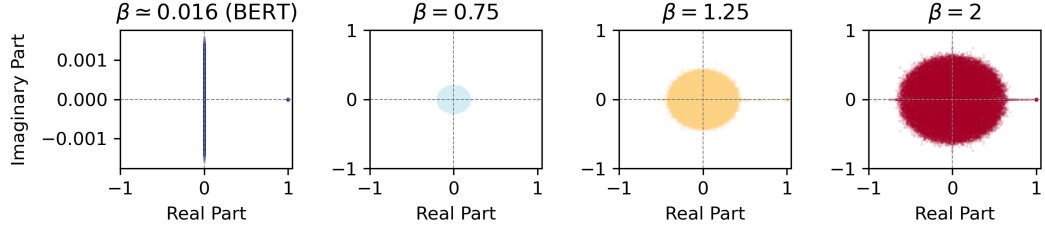

Figure 7: Spectrum of a $512 \times 512$ self-attention matrix for various values of the query/key variance parameter $\beta$.

### C.3 ENTROPY COLLAPSE CAN BE MITIGATED BY LOW LEARNING RATE

Figure 8 is obtained with the same set-up as fig. 3 but with smaller learning rate (5e-4→1e-4).

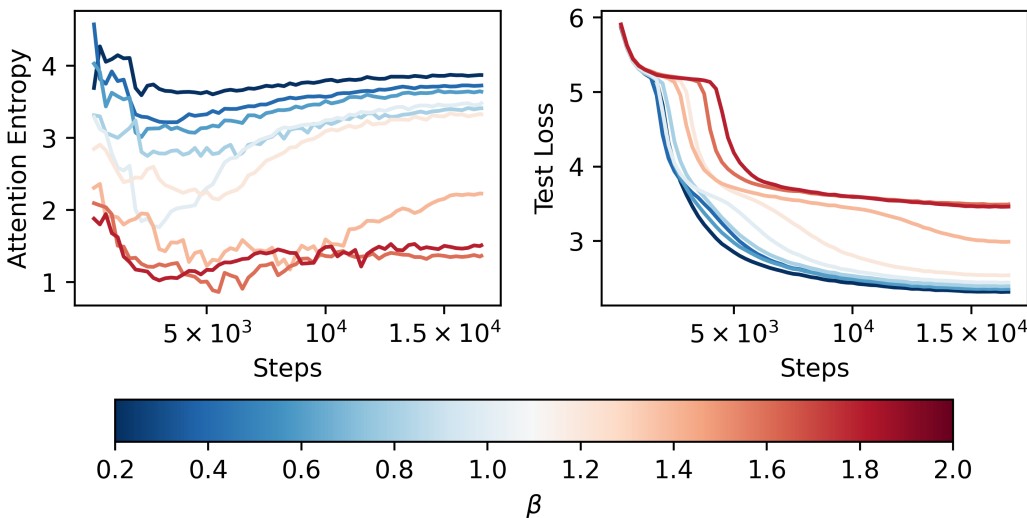

Figure 8: Entropy collapse can be partially mitigated by smaller learning rates.

### C.4 PHASE TRANSITION IN INFINITELY DEEP TRANSFORMERS

A natural question is whether signal propagation can remain stable at infinite depth. While this is not possible with ReLU activations, it becomes a genuine phenomenon when using tanh. This behaviour was first observed by Poole et al. (2016) in MLPs, and more recently extended to Transformers by Cowsik et al. (2024). In fig. 9, we confirm the phase transition in forward signal propagation, but emphasise that achieving this behaviour requires placing the MLP in the chaotic phase, which is associated with exploding gradients and unstable training.

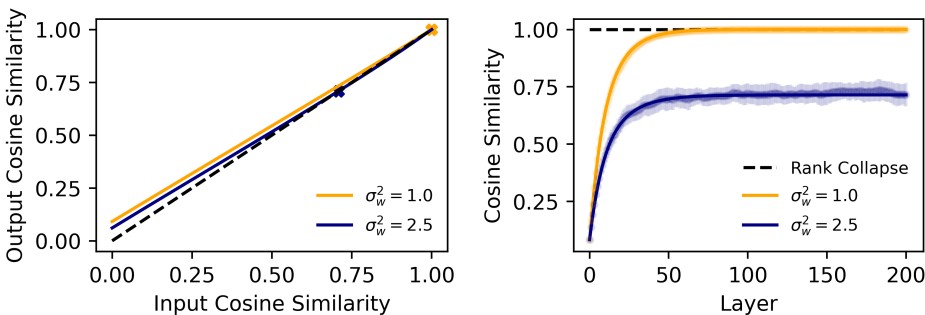

Figure 9: **Phase Transition to Infinitely Deep Signal Propagation.** *(Left)* Cosine-similarity update map of a full transformer block with `tanh` activations in the MLPs. By tuning the MLP variance to enter the chaotic regime, the collapsing effect of self-attention can be counterbalanced, resulting in a non-trivial fixed point in the similarity dynamics. *(Right)* Iterating the update map reveals the evolution of cosine similarity with depth—predictions align closely with experimental observations.

**(Left)** Theoretical predictions for a full Transformer block initialized in the low query/key variance regime, with MLP weights initialized as $\sigma_w = 1.0, 2.5$, $\sigma_b^2 = 0.1$. Residual connection scaling factors are set to $\alpha_{\text{SA}} = 6.0$ and $\alpha_{\text{MLP}} = 1.0$.

**(Right)** Evolution of the overlap by iteratively applying the map from the left panel over 200 layers, starting from an initial overlap of $\mathcal{O}(d^{-1/2})$. Solid lines denote theoretical predictions, while dots represent experimental results averaged over 25 random initializations, each evaluated on 10 input sequences (error bars are the standard deviation).

## C.5 TRAINING DYNAMICS FOR THE EXPERIMENTS IN THE MAIN TEXT

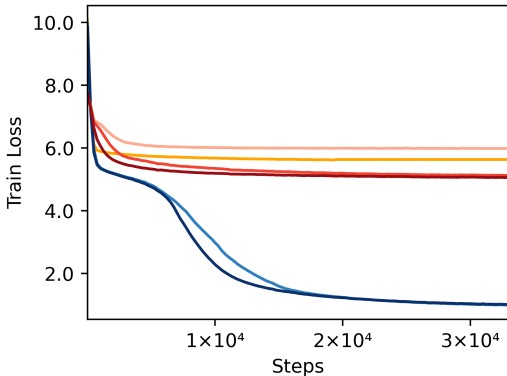

Figure 10: Training loss corresponding to the experiment shown in fig. 1e.

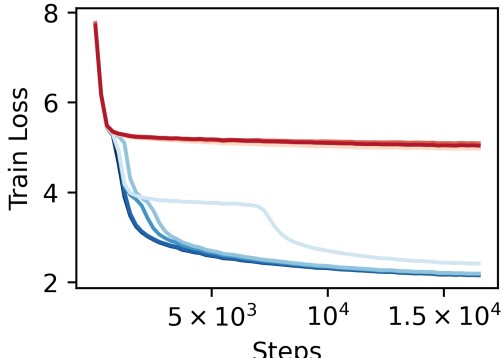

Figure 11: Training loss corresponding to the experiment shown in fig. 3b.

## C.6 AUTOREGRESSIVE MODELS

All experiments were performed on a 50-layer autoregressive transformer with post-normalization. We used hidden size 720, intermediate (MLP) size 720, and `ReLU` activations. The weight and bias variances were set to $\sigma_w^2 = \sigma_v^2 = 0.2$ and $\sigma_b^2 = 4 \times 10^{-4}$, respectively. Both the self-attention and MLP blocks employed scaling coefficients $\alpha_{\mathrm{SA}} = 1.0$ and $\alpha_{\mathrm{MLP}} = 1.0$. Results were averaged over 10 independent model initializations and 10 sequences of length approximately 200 tokens.

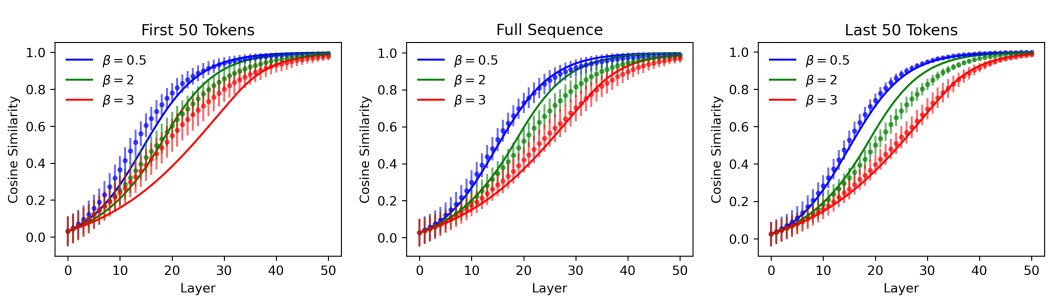

Figure 12: **Propagation in Autoregressive Models.** Cosine similarity between token representations as a function of layer depth for three values of $\beta$. The solid lines show our theoretical predictions, while the markers with error bars show empirical measurements obtained by averaging over all token pairs, across 10 sequences of length $\approx 200$ and 10 independent model initialisations. *Left:* For the first 50 tokens, the simplex geometry is not yet established, and the empirical evolution departs from the two-parameter theory. *Middle:* Averaging over the full sequence partially restores the structure but still mixes early and late tokens. *Right:* For the last 50 tokens, the geometry is well approximated by our theory: error bars are small, indicating strong concentration of cosine similarity, and the empirical curves follow the theoretical predictions. This illustrates that in causal models the simplex structure—and hence rank-collapse dynamics—emerge progressively and are captured accurately only for sufficiently late tokens.

