# OpenReview forum: "Two failure modes of deep transformers and how to avoid them: a unified theory of signal propagation at initialisation"
_ICLR.cc/2026/Conference — ICLR 2026 Poster_

### Official Review · Reviewer_5iSN · 2025-10-24

**Soundness:** 3
**Presentation:** 3
**Contribution:** 4
**Rating:** 8
**Confidence:** 4

**Summary:**

The paper analyses the signal propagation, i.e. the evolution of similarities between token embeddings in the self-attention layer and other layers of the transformer architecture. The paper focuses on the infinite context length limit (while also discussing the finite number approximations), or regimes where the context length is very large. They compute the correct asymptotic context length-dependent initialization scaling of the attention parameters that makes the limit non-trivial and also compute the concrete multiplication-constant threshold under which the attention matrix converges to uniform attention, causing rank collapse and above which the attention matrix converges to entropy-collapsed one where tokens only focus on finite number of other tokens despite asymptotic context length. Both these regimes constitute trainability issues. The authors further analyze the size of the gradient and show that it is vanishing in the small initialization regime. The authors then analyze the strength of the residual connection, which can alleviate the rank collapse in the small initialization regime. The authors obtain concrete algorithm to determine the trainability for given hyperparameters, which is then used to construct trainability diagrams. Finally, authors experimentally confirm their theory and evaluate some methods to boost trainability.

**Strengths:**

-	S1: The theoretical contribution of the paper is quite strong and, to the best of my knowledge, novel. Regarding novelty: while there are some pieces of the puzzle scattered around the related works, none provides such quantitative characterization, especially in the infinite context length limit or large contexts. None of the works unifies rank and entropy collapse regimes with precise threshold that divides them. The paper also convincingly discusses the non-asymptotic deviations, provides precise gradient norm analysis (novel) and connects it with the rest of the transformer (partially novel). Regarding the strength, I think the quantitative nature of the results, reasonable scaling regime and the potential for practical use make the paper useful for both theoreticians and practitioners.
-	S2: Besides being mostly novel, the authors seem to also accurately contextualize their novelty respective to the related work. I manually checked the related work (and also used AI to double-check) and it seem the authors correctly cited the literature in places where their results built on or aligned with some of the previous results.
-	S3: The paper is mostly quite clear and fairly easy to read.

**Weaknesses:**

-	W1: Despite general clarity, at times the paper would benefit from more detailed discussions. For instance, it should be discussed in the main body, why do the results require IID embeddings. If I guessed it correctly from the Appendix it seems that only to guarantee some geometric properties in the first layer. But this should be discussed because if this is not the case, then it is not clear how the IID property applies in the intermediate layers. Another example is that terms used in Equation 7 are only defined in Equation 9. Some statements are a bit vague and not justified-enough. In particular the discussion in lines 401-405 seems to reflect authors’ intuition, but I don’t see why this should be the correct explanation. Furthermore, Figure 1(e) would benefit a legend. I raise a couple more unclear points in the questions section.
-	W2: One peculiar decision is to scale up the residual connections instead of the output of the modules (for instance value matrix or MLP). As far as I am concerned, this doesn’t provide any benefit compared to the other option and, to the contrary, in the case of pre-norm LN causes exponential growth of embeddings, which is a big issue because the contributions of attention layers and MLPs will diminish at initialization as we go deeper through the network (plus, numerical instability might be an issue at large depths).
-	W3: The paper could go a bit more into depth about how to use Algorithm 1 to effectively determine the correct per-layer $\beta$ values in practice.

Typos:

-	l125 “applicaitons”

-	l141 fig 1c

-	l228 “initialise”

-	l319 the $\tau$ should be indexed

**Questions:**

-	Q1: To what extent are your proofs inspired by the either of the cited works [1,2,3] or some other works?
-	Q2: You haven’t disclosed LLM usage. I assume this means you haven’t used LLMs beyond some text adjustments and grammar checks. Is that right?
-	Q3: In Figure 1e, would the red and orange figures eventually decrease all the way down to blue curves or do you think they are stuck in local minima?
-	Q4: The normal distribution assumed in line 229 of the attention scores is also enforced in practice? If not, can it interfere with your theory because you are potentially creating heavier-tailed distributions?
-	Q5: In Figure 3(a), why does the entropy for the red curves actually go down? I understand they start lower than the blue curves, which is predicted by your theory, but why go down? As you show in the Appendix, smaller learning rate can alleviate this. Is it only an optimization issue then? Could LR warm-up help?
-	Q6: In result 2, you normalize with $T/log(T)$. This basically means that per-weight gradient converges to zero at an inverse rate. Does this mean we get vanishing gradients even in the optimal case?
-	Q7: What can be said in result 1 when $\beta=\beta_c$?
-	Q8: Can you elaborate on point (1) in line 396? How can we see this statement?
-	Q9: Can you elaborate on your discussion in lines 401-405? How can we see that this actually explains the phenomenon?
-	Q10: How do your results adjust in the case of multi-head attention?
-	Q11: More importantly, what can we learn about causal attention? It seems to me that causal attention cannot be solved by initialization and your results sort-of show it. Is that right? What would you recommend practitioners instead?
-	Q12: In line 463 you say that layer normalization is crucial to avoid entropy collapse. Where is this shown in your work?

[1] Carlo Lucibello and Marc Mézard. Exponential capacity of dense associative memories. Physical
Review Letters, 132(7):077301, 2024.

[2] Alireza Naderi, Thiziri Nait Saada, and Jared Tanner. Mind the gap: a spectral analysis of rank
collapse and signal propagation in transformers. arXiv preprint arXiv:2410.07799, 2024.


[3] Lorenzo Noci, Sotiris Anagnostidis, Luca Biggio, Antonio Orvieto, Sidak Pal Singh, and Aurelien
Lucchi. Signal propagation in transformers: Theoretical perspectives and the role of rank collapse,
2022. URL https://arxiv.org/abs/2206.03126.

**Summary:**
I consider the paper to have strong and novel contribution. While there are some aspects that can be improved and some questions that deserve answering, I think the paper should at least be accepted. Based on the discussion period and other reviewers’ points, I will then try to refine my judgement so as whether to recommend also spotlighting.

---

> ### Author Response · Authors · 2025-11-21
> **1/2**
>
> We genuinely thank the reviewer for appreciating our work and for carefully checking the related work.
>
> We address the weaknesses and the questions in the following two comments.
>
> ---
>
> # Weaknesses
>
> ### W1
>
> > why do the results require IID embeddings. [...] it is not clear how the IID property applies in the intermediate layers.
>
> We fully agree that this should have been made clearer, so we added two paragraphs (one in Section 2 and one in Section 3.2) to explain the geometric properties required to derive our results.
>
> Briefly, at initialisation the embeddings are i.i.d. by default, which yields a simplex geometry with $(q = 1, p = 0)$. Although the i.i.d. property breaks down after the first layer, working in the infinite–sequence-length limit ensures that the simplex geometry is preserved across depth, but with updated values of $q$ and $p$ (in particular, $p > 0$ once tokens become correlated).
>
> The machinery we use to track the evolution of these two geometric order parameters is therefore not the standard REM (which assumes independent Gaussian variables), but a modified version that we developed specifically to handle this correlated geometry.
>
> We hope that the added paragraphs help clarify this framework.
>
>
> > terms used in Equation 7 are only defined in Equation 9.
>
> We apologize for this, and it has been corrected, together with the other typos the reviewer identified in the new version. Thanks again for your contribution.
>
> > Figure 1(e) would benefit a legend.
>
> Thank you for pointing this out, we have added it.
>
> We address the remaining issues in the relevant questions.
>
> ### W2
>
> This is an interesting point. Although we did not focus on this specific effect, our theory can also accommodate the alternative modification proposed by the reviewer, which should now be clearer with the expanded Section 4. We concentrated on the strength of the skip connections because this approach is more common in the literature. Indeed, the solution that naturally emerges from our theory for the issue you correctly identify is to use *gain-controlled* attention instead, which resolves the propagation problems of self-attention without modifying the strength of the residual connections.
>
>
> ### W3
>
> We absolutely agree, and this point was also raised by another reviewer.
> The short answer is that one should use the algorithms we propose to check at each layer what the critical $\beta_c$ is, which depends on the geometric quantities at that layer.
> In the experiments in the paper, we used BERT as a case study, particularly in Fig. 1. This model uses post-norm, making $q=1$ at the input of self-attention layers at any depth. Since we know that $p$ increases with depth, the associated critical threshold also increases due to its dependency on $p$. So in this case, choosing $\beta$ to avoid entropy collapse at the first layer is enough to avoid it at all layers.
> But for general Transformer variants, one should be careful and choose $\beta$ to avoid entropy collapse at all layers.
>
> ---
>
> # Questions
>
> - Q1: While we share with [1] the observation that self-attention resembles the REM, our focus is quite different and independent. We use this resemblance to study signal propagation with *correlated* token embeddings, and one of our main contributions is precisely to handle this more general correlated simplex geometry of tokens, from which Result 1 directly follows. The interpretations are also orthogonal: in our case, the softmax sum runs over the sequence length, whereas in their case it runs over stored (independent) patterns.
> Deriving Result 2 builds on the vanishing-gradient proof from [3], but our result is more general: instead of assuming uniform attention, we allow the Inverse Participation Ratio (IPR) to enter explicitly and make use of our Result 1.
> Regarding the idea of modifying the attention mechanism by removing the mean value, as we note in the main text, this was proposed in [4] and later in [2]. Our framework confirms that, from a signal-propagation perspective, this is indeed an interesting and effective modification—one for which a theoretical guarantee was previously missing. Reference [4] also moves in this direction, but in a setting that does not include LayerNorm, which in light of our results is crucial for avoiding entropy collapse (see Q12).
>
> [4] Noci et al. "The shaped transformer: Attention models in the infinite depth-and-width limit." NeurIPS 2023

---

> ### Author Response · Authors · 2025-11-21
> **2/2**
>
> - Q2: Yes, we have *not* used LLMs beyond some text adjustments and grammar checks.
>
> - Q3: We think that initialising in the entropy collapse phase causes a problem which is of an optimisation nature. Indeed, we believe that eventually the network should be able to unlearn the random saturated patterns and learn semantic ones. But we have no idea on which time scale this would happen.
>
> - Q4: Gaussianity is not enforced directly in practice, but it's the result of the central limit theorem in the scalar product between query and key. So everything is consistent between theory and practice (as long as one doesn't initialise weights with heavy-tailed distributions).
>
> - Q5: Our Result 2 highlights the role of the participation ratio in the norm of query/key updates. Since only the already large queries/keys are the ones that participate, we think one can conclude that they're also the only ones to receive a gradient. This makes them even larger, i.e., reinforces the random patterns, causing entropy to go down even further.
> So we agree with you that LR warmup would help in this case (even if applied only to query/keys).
>
> - Q6: The result holds only for the first gradient, so we believe that once the simplex geometry is broken enough by training, the scaling may become different.
>
> - Q7: Since the IPR is a continuous (but not differentiable) function, it still holds that $\text{IPR}=0$ in that case, so the behavior is described by the low $\beta$ phase.
>
> - Q8: In Figure 3c, we plot the Frobenius norm of the query gradient (obtained with PyTorch) of the first layer of different Transformers at varying embedding dimensions and sequence lengths. We do this for different $\beta$ values. If one multiplies them by the quantity $T/(d^2 \sqrt{\log T})$ which we identify in Result 2, then the curves collapse onto the same curve which is independent of $d$ and $T$. This confirms we identified the right scaling. Does this answer the question, or did you mean something different?
>
> - Q9: This is an intuition that tries to unify Fig. 3c and Fig. 3b: In Fig. 3c we see that we have vanishing query gradients in the small $\beta$ phase, but in Fig. 3b we see that the low $\beta$ phase is the only one that effectively trains. Our intuition is built on the fact that even though one has vanishing query gradients, the rest of the Transformer components still receive a gradient thanks to the skip connections. So, once the simplex geometry is broken enough (e.g., semantic embeddings are learnt), Result 2 breaks down, and we know this is the case because in Fig. 3b we are effectively training.
>
> - Q10: The adjustment is straightforward: the result holds for each head individually. The only change needed is to incorporate the effect of the output layer, which is just a linear transformation. For instance, if this layer is initialised with variance \(\sigma^2 = d^{-1}\), then the resulting behavior is identical to what we have described.
>
>
> - Q11: While in causal attention rank collapse does not occur at the level of the full sequence, collapse in the representations of tokens near the end of the sequence could still happen through a mechanism similar to the one we consider here. We think (though further experiments are necessary) that removing the average Value—the “sequence norm” operation of gain-controlled attention—could also benefit causal models, especially when processing very long sequences.
>
>
> - Q12: We agree this conclusion was rushed; we have now expanded it in the new version. We meant to say that the critical threshold we identify in Result 1 depends inversely proportional to the average norm. If one never normalises, the average norm grows with depth (mainly due to the skip connections), making the critical threshold smaller and smaller with depth and so eventually causing entropy collapse.
>
> ---
>
> Thank you for the very interesting questions and comments, to which we hope we have addressed as deeply as the reviewer was expecting. If something is unclear or deserves more explanation, we are happy to elaborate further.

---

> > ### Comment · Reviewer_5iSN · 2025-11-21
> >
> > Thank you for your detailed explanations. I consider many of the points solved now. I will follow-up on the points that require further discussion. Before that, however, could you please highlight the changes you made to the paper with a different color? It will be more convenient to compare.
> >
> > - **Regarding W1:** I understand your explanation, but does this mean that you could arrive at the same results by just assuming that the embeddings in a first layer follow the simplex geometry?
> >
> > - **Regarding W3:** Is it enough to avoid the entropy collapse? Do you claim that one does not need to avoid the rank collapse? From your experiments this seems to be the case but I didn't have an impression that you would try to narrate such a message. In the case of more complicated architectures -- if one wants to set the initializations right, does this mean solving some fairly complex multi-dimensional (one parameter per each layer) optimization problem?
> >
> >  - **Regarding Q4:** But the products of gaussian variables do have heavier tails than gaussian variables, so this makes the sum also have heavy tail (we know that CLT doesn't work for tails in finite sample).
> >
> > - **Regarding Q5:** Why should the gradients make them larger though? The gradients can also try to undo them by making them smaller. In fact this is a more intuitive guess since the initial attention weights don't have semantic meaning.
> >
> > - **Regarding Q6:** But if the gradient is vanishing even in the optimal case, than perhaps the "broken enough" will take a lot of training steps.

---

> ### Author Response · Authors · 2025-11-23
>
> We thank the reviewer for the follow-up. Thank you also for suggesting to upload a PDF with highlighted differences; we have done so and hope this makes tracking the changes easier.
>
> **Regarding W1:**
> Yes, that is exactly the case.
>
> **Regarding W3:**
> It is crucial to avoid both entropy collapse and rank collapse. Our point was that adjusting $\beta$ primarily helps prevent entropy collapse. For successful training, one should avoid entropy collapse at *all* layers by tuning $\beta$, and separately adjust other parameters --- e.g., the strength of skip connections, or, as you suggest, the variance of appropriate linear layers -- to prevent the average cosine similarity $\rho$ from saturating to 1 before the desired network depth. For example, the blue curves in Figure 1e are the only ones that avoid *both* issues (the yellow curve suffers from rank collapse with $\rho \simeq 1$, and the red curves from entropy collapse).
>
> Because these effects are rather distinct to control, finding the right initialization is not a complex multi-dimensional optimization problem. One can proceed sequentially: start by choosing a low initial $\beta$ (say 0.1) and inspect how the cosine similarity evolves with depth (as in Figure 1d). If $\rho$ saturates to 1, adjust the skip-connection strength (or the alternative you suggest). Once that issue is resolved, check that the critical threshold is not crossed at any layer. If it is, fix $\beta$ at the relevant layers and re-check that cosine similarity does not saturate.
>
> **Regarding Q4:**
> The only quantity that matters for our argument is the resulting *variance* of the attention scores. So while you are correct that products of Gaussian variables have heavier tails, all we need is that the scores have finite variance—which is guaranteed as long as query/key weights are initialized with finite variance.
>
> **Regarding Q5:**
> Yes, we fully agree. Gradients could indeed try to undo them; however, the issue arises when the learning rate is too large: in that case, an update may increase the norm of the query/key weights, effectively enlarging $\beta$ and pushing the system toward entropy collapse. This also aligns with your intuition that learning-rate warmup helps: early updates should affect the weight norms as little as possible, allowing semantic structure to form gradually without triggering entropy collapse through sudden norm growth.
>
> **Regarding Q6:**
> We see your point. Assuming the scaling identified in Result 2 is correct, each gradient component is $O(T^{-1/2})$. Thus, choosing appropriate (different) learning rates for the query/key weights and the rest of the network should suffice, and this is precisely what Adam does. So one should indeed expect Adam to overcome this scaling, especially for large (but finite) sequences.
>
> We hope this clarifies the remaining points, and we remain at your disposal for any further questions. Thank you again for the interesting and thoughtful remarks.

---

> > ### Comment · Reviewer_5iSN · 2025-11-24
> >
> > Thank you in engaging in an interesting discussion. Only two of the above points require further discussion.
> >
> > - **W3:** To me, the procedure you described sounds quite complicated. You might need to do a lot of back-and-forths, because setting $\beta$ value at any layer will influence all the subsequent layers.
> >
> > - **Q4:** If I understood it correctly, you also need to use SLLN (and CLT for finite-sample) on the exponentials of the scores, for which the tails of the scores matter. Please let me know if I misunderstand this part of argument.

---

> > > ### Author Response · Authors · 2025-11-26
> > >
> > > Thank you for continuing the discussion.
> > >
> > > - **W3:** We fully see your point, but we believe two simplifications make the procedure smoother than it may initially appear. First, in eq. 14 (of the new version) the action of the self-attention layer is:
> > >   $$
> > >   q_{\mathcal{S}} = \sigma_v^2 \left( p + (q - p)\, Y^{(2)}(\beta, q, p) \right),
> > >   \qquad
> > >   p_{\mathcal{S}} = \sigma_v^2 p ,
> > >   $$
> > >   and we recall from **Result 1** that
> > >   $$
> > >   Y^{(2)}(\beta, q, p) = 0 \quad \forall\, \beta < \beta_c(q,p).
> > >   $$
> > >   This means that forward propagation is *not* affected by changes in $\beta$ as long as the critical threshold is not exceeded. This leaves room to adjust the other parameters before the values of $\beta$ start influencing propagation.
> > >
> > >   A second simplification comes from how the critical threshold depends on similarity:
> > >   $$
> > >   \beta_c(q,p) = \sqrt{\frac{2}{q\,(q - p)}}.
> > >   $$
> > >   Thus the threshold *increases* as similarity increases. Since similarity tends to increase with depth, entropy collapse is primarily a risk for the *earlier* layers (assuming the use of layer normalization).
> > >
> > >   In summary: starting with a small enough $\beta$ so that the first layers lie below the threshold is usually sufficient to avoid entropy collapse. One can then adjust the other parameters to prevent rank collapse as well. Because forward propagation remains unaffected by $\beta$ as long as the threshold is not crossed, one has flexibility to tune those parameters independently of the exact $\beta$ values.
> > >
> > > - **Q4:** Thank you for the clarification - and apologies for the earlier confusion. This is a very interesting point. The result indeed depends on the tails of the distribution of the attention scores, as you correctly point out; in particular, it depends on the exponent $\rho$ in the tail behaviour $P(X > x) \sim e^{-x^{\rho}}$. You are absolutely right that the critical threshold we state - which depends on this exponent - is the correct one only when the embedding dimension $d \to \infty$; for finite $d$, the tail behaviour does matter. We thank the reviewer for identifying such a subtle point, and we agree that we should make this explicit (namely, that we first take $d \to \infty$ and then $T \to \infty$) when stating Result 1.
> > > In practice, this dependence does not appear to significantly affect the behavior we observe. The finite-size effects arising from finite $T$ seem to be the dominant factor, and the resulting predictions for the cross-over region match experiments closely.
> > >
> > > We hope this clears up the remaining point and thank you again for the interesting points you brought up. We remain at your disposal for further discussion.

---

> > > > ### Comment · Reviewer_5iSN · 2025-11-26
> > > >
> > > > Thank you for clarifying both points. I now find all the points of our discussion to be resolved.
> > > >
> > > > I will reevaluate my scoring once I read in detail the other reviewers' discussions too.

---

> > > > > ### Author Response · Authors · 2025-11-26
> > > > >
> > > > > That's wonderful to hear, thank you for taking the time to engage in such a thoughtful and thorough discussion.

---

### Official Review · Reviewer_1xRi · 2025-10-30

**Soundness:** 3
**Presentation:** 3
**Contribution:** 3
**Rating:** 6
**Confidence:** 3

**Summary:**

The authors propose to use the REM to provide an asymptotic characterization of signal propagation in Transformer (self-attention layer, skip connections, Layer Norm, MLP) and explain how different initialization scalings cause rank and entropy collapse.

**Strengths:**

- The paper provides a precise asymptotic analysis on the evolution of the average cosine similarity and the average IRP with respect to the initialization scaling.

**Weaknesses:**

- There are many unclear arguments.
    - Why is the variance of the attention scores $\sigma_a^2$? (Doesn't it depend on the value of the variance of $X_t$?)
    - Sec B.1 L666 says $\sigma_Q^2=\sigma_K^2=\sigma_a^{\color{red}2}/d$ which is different from eq (8). Also the conclusion in (15), $Cov(a_{ts}a_{\tau\sigma})=\sigma_a^2q_{ts}q_{s\sigma}$ is $d^2$ times larger than eq (7).
    - Why does the attention score variance should $O(\log T)$?

- The asymptotic prediction (Result 1) does not perfectly match the empirical results. The authors said that this is because $T$ is finite. How about the nonasymptotic results with concentration bounds? What happens if we increase $T>1024$?

- Why $\tilde \beta_c=\beta_c/2$ does not act as a critical value in Fig 3?

- Violation of the iid assumption means that (13) may not hold, but it does not answer why gradient does not vanish with the skip connection. Why do we have nonvanishing gradients when $\beta<\beta_c$ with skip connection? Also it seems like the iid assumption is violated during training, even without skip connection, so we may not conclude that $\beta<\beta_c$ implies vanishing gradients.

**Questions:**

see weaknesses

---

> ### Author Response · Authors · 2025-11-21
>
> We thank the reviewer for bringing up several important points to clarify.
> We proceed by addressing the weaknesses directly.
>
> ---
>
> ### Clarifications
>
> - Regarding the variance of the attention scores, it is exactly the one the reviewer also wrote:
> $$
> \mathrm{Cov}(a_{ts}, a_{\tau \sigma}) = \sigma_a^2  q_{t\tau}  q_{s\sigma},
> $$
> where $\sigma_a^2$ is the geometry–independent term in the correlation between attention scores, and not directly the variance itself.
> - We apologise for the typo in the appendix: the correct formula is the one in the main text,
> $\sigma_Q^2 = \sigma_K^2 = \sigma_a / d$.
> - Concerning the $\mathcal{O}(\log T)$ scaling, Section 3.1 is dedicated to supporting the parallel between self-attention and the Random Energy Model (REM). The REM suggests that this is the exact scaling needed to obtain a sharp transition between the spread (rank–collapse) regime and the localised (entropy–collapse) regime.
>
> ### Finite-size effects
>
> Our theory is asymptotic in the limit of sequence length $T$, but we also derive the finite-size corrections explicitly. From this analysis, we find that the asymptotic predictions remain exact for finite $T$ whenever $\beta < \tilde{\beta}_c$ or $\beta > \beta_c$. The region in between should be interpreted as a crossover between the two asymptotic regimes, exhibiting mixed features of both phases. As $T$ increases, these finite-size effects progressively diminish—for example, in Figure 2 the empirical diagram would more closely resemble the asymptotic one.
>
>
> In Figure 3(a,b), $\tilde{\beta}_c = \beta_c / 2$ indeed behaves as an effective critical value: it marks the onset of large-$\beta$ effects. In that setting, $\beta_c = \sqrt{2}$ (so $\tilde{\beta}_c \approx 0.7$), and the curve for $\beta = 1$ shows precisely this mixed behaviour: the initial lower entropy causes a significantly slower learning than models with smaller $\beta$.
>
>
> ### Vanishing gradients
>
> Only the gradients of the query/key projections vanish.  The rest of the components of the Transformer, at all layers, still receive gradients thanks to the skip connections. Even though our theory concerns only the first backward pass, we can still conclude the following: without skip connections, gradients would not propagate, and both the attention mechanism and the rest of the Transformer would remain essentially at their initialisation values. This is exactly the argument we use to assess the trainability of deep networks in Fig. 1.
>
> ---
>
> We thank the reviewer again for spotting important typos and for thoughtfully engaging with our work. We believe the weaknesses raised have now been addressed in this reply and in the revised manuscript, and we would be grateful if the reviewer would consider reflecting this in your evaluation. If any issue remains unclear, we are happy to provide further clarification.

---

### Official Review · Reviewer_X3PM · 2025-10-31

**Soundness:** 3
**Presentation:** 2
**Contribution:** 3
**Rating:** 6
**Confidence:** 3

**Summary:**

The paper conducts a unifying analysis of two failure modes in signal propagation in Transformer models, namely the rank and entropy collapse, under the assumption of infinite width and sequence length. The authors use a known mapping of the self attention matrix to a Random Energy Model (ERM) [1], which allows them to conduct analysis that, contrary to the previous work, neither assumes uniform attention scores, nor taking expectations in the numerator and denominator of the self-attention separately.
Following that they propose a new sequence-length dependent initialization scheme for the query and key weights, and derive the formula for the evolution of the average cosine token similarity as a function of the sequence-length independent initialization variance and the input embedding correlation. They also extend our understanding of the vanishing gradient problem beyond the uniform attention assumption, and provide an analysis of a full Transformer block, which lets them construct cosine-similarity evolution diagrams also for full Transformers. They verify their theoretical results with experiments on a BERT-style Transformer models.

[1] Carlo Lucibello and Marc Mézard. Exponential capacity of dense associative memories. Physical Review Letters, 132(7):077301, 2024.

**Strengths:**

1. Unifying view of two failure modes of attention is novel and valuable for the signal propagation community. I think that working under less strict assumptions than previous work (uniform attention, separate treatment of nominator and denominator in attention) is valuable.
2. Using ERM in the context of signal propagation analysis is novel and original.
3. The part of the paper that focuses on a single self-attention layer is good quality and well presented. It’s commendable that the authors extend their analysis to a full Transformer block, and provide experiments for deep Transformer models.

**Weaknesses:**

I’m open to increasing my score if the authors address the following weaknesses:

1. The analysis of the average cosine similarity while passing through the full Transformer block (Section 4) is rushed. I believe some other parts could be shortened (like the setup in Section 2) in order to make space for a more thorough analysis of the full Transformer block. The result from eq. 14 is not interpreted and authors do not clearly walk the reader through how they arrive at Algorithm 1 from it. I believe this type of analysis is valuable but should be presented better in the main text of the paper.
2. The analysis of the propagation in terms of $\beta$ and $\beta_c$ seems to be done only for a single layer Transformer. I would be curious to see some discussion and some experiments similar to what’s presented in Figure 3d on how it behaves with increased depth. Especially, as the paper already discusses deeper models, but focuses only on the residual scaling $\alpha_{SA}$.
3. The experiments concern only BERT-style architecture. I believe the authors should provide some empirical results and comment on how their theoretical result and proposed algorithms for obtaining trainability diagrams behave in architectures with causal masking, like GPT-style autoregressive Transformer. These architecture is arguably more relevant for the DL community these days.
4. There is no discussion about the gradients in the entropy-collapse regime. Section 3.3 just discusses the gradient formula for $\beta < \beta_c$. I would be interested in seeing if there is a way to explain with gradients how it happens that for large $\beta > \beta_c$ we get stuck in the entropy collapse regime.
5. Regarding figures: 1) Figure 3c does not have a legend. 2) I would appreciate seeing also the training loss curves (in the appendix) for the experiments for which the evaluation/validation loss is presented.

**Questions:**

1. Can you explain (briefly in the main text of the paper and maybe more in detail in the appendix) how one arrives at equations 7-15 from algorithm 1? As I understand it, it comes from previous work, but it would be useful to walk the reader through how one arrives at these formulas.
2. How does $\beta_c$ depend on the model depth, assuming models with and without layer normalization? How does $\beta_c$ depend on the layer number assuming a model with a fixed depth (with and without layer normalization)?
3. Can you produce a trainability diagram in the deep transformer case like the one in Figure 1d, just for varying values of $\beta$?
4. Is the entropy collapse regime also associated with a vanishing gradient problem? From fig. 1e it seems that models in this regime quickly stop learning. If the whole norm of the gradient in eq. 13 does not disappear, maybe the gradients wrt to just some subset of query and key parameter do? Do the attention scores stay roughly the same (with the same few large scores) during training or do they change between many low-entropy settings?
5. How does masking influence the results? Would a causal mask like in GPT-style attention change Result 1 and eq. 14? How would the phase diagram from fig. 2 look under attention with a causal mask?
6. In the paper $q_{tt}$ is referred to as the token norm (for example in line 295). Are you sure it is not a squared token norm?

Suggestions on writing:
* The sentences around eq 14. could better suggest that the equation describes propagation through a self-attention layer with skip-connection. From the section title I was expecting a single equation describing the full Transformer block, and the sentence leading to the equation just mentions skip-connection.
* Lack of consistency with capitalising Transformer/transformer (lines 14 and 42 for example)
* Broken sentence in line 113 (the strength the scale)
* Line 131: „highly highly”
* While I appreciate the explanation in appendix B.1, the writing there could be improved (take a look at the second sentence in line 652).
* $q_{ts}$ is used in line 233 before its defined way later in the paper

---

> ### Author Response · Authors · 2025-11-21
>
> We appreciate that the reviewer recognises the novelty of our work and its contribution to the signal-propagation literature. Because the weaknesses and questions raised are closely related, we address them together below.
>
> ### Full Transformer Block (W1, Q1)
> We agree that Section 4 was too brief. In the revised version that we have already uploaded, we expanded the derivation of Eq. 14, added the relevant signal propagation theory for MLPs of prior works, and detailed the treatment of residual connections and layer normalisation. This should make the connection to Algorithm 1 much clearer.
>
>
> ### Critical $\beta$, Depth, and LayerNorm (W2, Q2–3)
>
> **About depth:**
>
> The theoretical analysis in terms of $\beta$ applies to self-attention layers at any depth. In the infinite-sequence limit, token representations follow a simplex-like random geometry parameterised by $(q, p)$, and this structure is preserved across layers, with $(q, p)$ evolving through depth. Result 1 yields a critical threshold
> $$
> \beta_c = \sqrt{\frac{2}{q(q - p)}},
> $$
> which depends on depth through the geometric quantities at that layer.
>
> **About LN:**
>
> LN affects the results because it rescales $(q, p)$ to $(1, p/q)$ at the point where it is applied, thereby altering the resulting critical threshold $\beta_c$. We emphasise that LN is essential in light of our findings: without it, $q$ grows with depth, causing $\beta_c$ to shrink and leading to entropy collapse for progressively smaller values of $\beta$.
>
> **About the diagrams:**
>
> In Figure 1, we consider a post-norm model, so the inputs to each self-attention layer satisfy $q = 1$. In this setting, the formula simplifies to $\beta_c = \sqrt{2/(1 - \rho)}$. Since $\rho = p/q$ increases with depth, the most restrictive condition occurs at the first layer, where $\rho \approx 0$. Thus, to avoid entropy collapse at all layers, it is sufficient to set $\beta < \sqrt{2}$. This is the criterion used to construct the diagram in Figure 1c.
>
> More generally, each layer should be initialised with a $\beta$ below its own $\beta_c$, computed from the corresponding $(q, p)$ at that depth.
>
> ### Gradients in the Entropy-Collapse Phase (W4, Q4)
> In the entropy-collapse phase, as you note, only a subset of queries and keys is effectively updated at the start of training. This is precisely what Result 2 formalises: the participation ratio appears in the gradient norm, showing that in this regime only a small set of large query/key components contributes to the updates.
>
> Thus, the issue is primarily an optimisation one: once these random patterns dominate the attention, the model struggles to “unlearn’’ them and replace them with semantic ones.
>
> This intuition is supported by two observations:
>
> - In Fig. 3a–b, the curve for $\beta = 1$ lies in a finite-size crossover region. After an initial entropy drop caused by reinforced random patterns, the model eventually learns semantic ones, though much slower than models with smaller $\beta$. Still, it learns—unlike larger-$\beta$ models, which fail to train within our experimental timescale.
>
>
> - Figure 2 of Zhai et al. shows the same effect: increasing $\beta$ at the beginning of training induces pathological entropy collapse, whereas increasing it later is far less harmful because semantic structure has already formed.
>
>
> Reference:
> Zhai et al. *“Stabilizing Transformer Training by Preventing Attention Entropy Collapse.”*
>
>
> ### Causal Masking (W3, Q5)
>
> Our results rely on the simplex-like geometry $(q,p)$ that emerges for long sequences. In causal models this geometry is violated for early tokens, but appears progressively for deeper ones. Hence signal propagation for the entire sequence cannot be captured by a few parameters, and rank collapse does not occur globally. However, collapse within representations of late-sequence tokens can still arise. We added a paragraph in Section 5 to expand on this. Thank you for the question.
>
> Thus, a plot like Fig. 2 would resemble the MLM case when restricted to later tokens, while the full-sequence geometry is more complex and cannot be described by just a couple of order parameters.
>
> ### Other comments
> (Q6) Thank you for pointing this out; it is indeed the squared norm. We have corrected this in the revision.
>
> (W5) We added the training-loss plots in Appendix D. We did not include a legend in Fig. 3c because all curves for the different $(d, T)$ combinations (listed in the caption) collapse onto one another, and a legend would clutter the figure.
>
> Thank you as well for identifying typos; all have now been corrected. We also made sure to define $q_{ts}$ before its first use (now defined at line 200).
>
> ---
>
> We thank the reviewer once again for the thoughtful and constructive feedback. We hope that our responses have addressed all of your concerns. Should this be the case, we would be grateful if you would consider reflecting this in your evaluation. If not, we are happy to provide any additional clarification.

---

### Official Review · Reviewer_zigt · 2025-10-31

**Soundness:** 4
**Presentation:** 2
**Contribution:** 2
**Rating:** 4
**Confidence:** 5

**Summary:**

This paper presents a unified analytical theory to explain and predict two failure modes of deep Transformer networks due to initialization: rank collapse and entropy collapse. The authors' central innovation is to establish a formal parallel between the self-attention mechanism and the Random Energy Model (REM) from statistical physics. This mapping allows them to analyze the behavior of self-attention as a phase transition. The authors present a parameter, $\beta$ (related to the variance of query/key weights), that functions like "inverse temperature" in the physics model. The theory finds that operating below a critical threshold ($\beta < \beta_c$) leads to rank collapse, while operating above it ($\beta > \beta_c$) leads to entropy collapse. The paper's practical solution is to operate in the rank-collapse regime (to avoid the more fatal entropy collapse) and use strong residual connections ($\alpha_{SA}$) to preserve signal diversity. The framework yields a simple algorithm to compute "trainability diagrams," which provide a principled and quantitative guide for selecting initialization hyperparameters (weight scale and residual strength) to ensure stable training. The authors' theoretical predictions are shown to closely match empirical measurements in BERT-style models.

**Strengths:**

The paper's strengths lie in its rigorous theoretical approach to a critical practical problem.

Strong Theoretical Analysis: The paper provides a comprehensive and asymptotically exact analytical framework. It goes beyond simple heuristics to derive precise update equations for token similarity (Result 1) and an exact expression for the gradient norm at initialization (Result 2). This allows it to analyze both forward signal propagation and backward gradient flow, providing a complete picture of the network's dynamics at initialization.
Novel Parallel with the Random Energy Model (REM): The analogy to the REM is the paper's key insight. By formally mapping the self-attention $\text{softmax}$ (Eq. 3) to the Boltzmann distribution (Eq. 6), the authors import a powerful analytical toolbox from statistical physics. This allows them to model the complex, random behavior of the attention layer and identify the "inverse temperature" parameter $\beta$ (Eq. 8) as the crucial knob controlling the system's behavior.
Effective Visualization of the Phase Transition: The REM analogy naturally leads to the discovery of a sharp phase transition at a critical threshold $\beta_c$. This core finding is visualized clearly and effectively. Figure 2 presents the theoretical phase diagram for a single layer, while Figure 1c translates this into a practical "trainability diagram". This diagram provides a clear map for clearly delineating the trainable zone from the Rank Collapse and Entropy Collapse zones.

**Weaknesses:**

My main concern is the lack of convincing experiments supporting the framework. While it is totally fine that the paper's focus is theoretical, it's a shame that there is no better evidence to support what is ultimately a very important practical problem - specifically when strong evidence would not be so difficult to provide. The core results are derived in the "limit of infinite sequence length." The authors acknowledge this creates "finite-size effects" and a "discrepancy between theory and simulations" (Fig. 2) in real-world. Also, the gradient analysis (Result 2) assumes token embeddings are i.i.d. (independently and identically distributed). The authors note this is only true at the moment of initialization ($t=0$) and is immediately broken by the very skip connections their solution relies on. Both these theoretical assumptions alone justify better experimental evidence for the hypothesis.

Furthermore, better support of how should the community benefit from more informed initialization - which is the whole point of the paper given the abstract - would be welcome.

Finally, I find the paper needs some restructuring and rewriting. Here are some pointers: Section 4 for instance is standing alone, and a bit odd with the rest of the paper. A standard "Related Works" section would be good, as opposed to simply a "Further Related Works" paragraph after stating the main paper contributions. A discussion section as a 3.2.3, followed by 3.3 The Backward Pass also reads rather strangely. You mention training a 20 layer BERT (line 472) in the Main manuscript, but in the supplementary material, it becomes a 30 layer BERT model (line 1214) - which is true? In addition, the paper is filled with typos...

**Questions:**

The authors map self-attention to the Random Energy Model, but REM assumes independent energies while attention scores are correlated. How do these correlations affect the validity of the phase transition predictions, especially for moderate sequence lengths where finite-size effects matter?

---

> ### Author Response · Authors · 2025-11-14
>
> We thank the reviewer for carefully reading our paper and for the constructive feedback.
> We also appreciate the reviewer’s positive remarks regarding our theoretical analysis, and we would be happy to engage in further discussion to better understand which empirical evidence they believe would most effectively strengthen the paper.
>
> We aimed to directly support our main theoretical findings as follows:
>
> **Regarding forward propagation:**
> - *Figure 2* supports our prediction for the evolution of cosine similarity after a self-attention layer.
> - *Figure 1d* compares our predicted evolution of the average cosine similarity through multiple full Transformer blocks with empirical results on BERT.
>
> **Regarding the backward pass:**
> - *Figure 3c* supports our prediction of vanishing gradients in the low-\(\beta\) phase.
>
> The remaining experiments (e.g., *Figures 1e* and *3a–b*) illustrate how the initialisation issues we identify affect the training dynamics of Transformers.
>
> ---
>
> We thank the reviewer for the detailed and constructive feedback. We address each of the raised weaknesses below.
>
> ### Finite-size effects and experimental support
>
> For finite system sizes, our theory should be interpreted as an exact description for $\beta < \beta_c / 2$ and $\beta > \beta_c$, with a crossover between the two regimes in the intermediate region. The experiments shown in *Figures 2* and *3c* are indeed influenced by finite-size effects (which we quantify in Appendix B.3), and they deviate from the asymptotic prediction only within this crossover region. This behaviour is consistent with theoretical expectations. We agree that expanding the discussion in the main paper will make this clearer, and we will take advantage of the additional space in the revision to that end.
>
> ### Gradient analysis and i.i.d. assumptions
>
> *Result 2* concerns the first gradient step, right at initialisation. While this result (supported empirically by *Figure 3c*) shows that query and key gradients vanish below a critical threshold, other components of the network still receive gradients through skip connections. These provide a pathway that bypasses the attention layers and therefore remains unaffected by the vanishing behaviour of the query/key pathway.
>
> To ensure we provide exactly what the reviewer finds missing, we would appreciate clarification: is the request for empirical confirmation that gradients continue to flow through the residual path despite vanishing query/key gradients, or for a different type of experiment? We are happy to include the relevant results.
>
> ### Practical relevance and informed initialisation
>
> Our goal in introducing the trainability diagram and initialisation algorithms is to show how the theory can guide practice. As noted by other reviewers, we will expand Section 4 to better connect our results with prior signal-propagation analyses for MLPs and expand on the role of depth in building the trainability diagrams.
>
> ### Paper structure, clarity, and typos
>
> We currently discuss related work directly in the introduction, but we agree that presenting this material in a dedicated **Related Work** section would make it easier to locate. In the revision, we will restructure this discussion accordingly and improve the transitions around Sections 3 and (expanded) 4.
>
> We also thank the reviewer for pointing out inconsistencies and typos. In particular, the correct number of layers in our BERT experiment is **30** (line 472), and we have corrected this mistake. We have gone through the manuscript again and revised all typos we identified, including those highlighted by the reviewers.
>
> ## Reply to "Questions"
>
> The question the reviewer poses is important and touches on the key geometric difference between our setting and the REM.
> After the embedding layer, tokens are represented by i.i.d. vectors, so they form a random simplex with $q = 1$ and $p = 0$.
> The central idea behind working in the infinite-sequence-length limit is that this simplex random geometry is preserved throughout Transformer components, with $q$ and $p$ varying in depth. In particular, the simplex geometry is preserved through the self-attention layer thanks to *Result 1* (Eq. 12), and through the MLPs because it is the same MLP that processes each token individually.
>
> Given this geometry, we derive our results (Appendix B.2) by studying a model that is more general than the REM. Incorporating correlations yields the critical inverse temperature
> $$
> \beta_c = \sqrt{\frac{2}{q(q - p)}},
> $$
> which depends on geometric quantities of the simplex. The standard REM is recovered by setting $q = 1$ and $p = 0$, which removes correlations among energies.
>
> ---
>
> We again thank the reviewer for the thoughtful feedback and remain at their disposal for further clarifications. We would be happy to understand which specific experiment they feel is missing and provide an updated version accordingly.

---

### Author Response · Authors · 2025-11-21

Dear AC,

Thank you for taking the time to oversee the reviewing of our paper. The main
contribution of our paper is a complete analysis of signal propagation across a
full Transformer block with self-attention, residual connections, LayerNorm, and
ReLU MLP, which yields an algorithm that accurately predicts the trainability of
deep Transformers.

We appreciate the additional effort of area chairing for papers that you have
only recently been assigned to. Below, we summarise the main points that emerged
from the reviews, and we detail how we improved the manuscript. We thank all
reviewers for their time, careful reading of the manuscript, and constructive
feedback, which helped us substantially improve the paper.

## Summary of reviews

Reviewers consistently praised the paper's strong and novel theoretical
contribution, highlighting its "rigorous theoretical approach to a critical
practical problem" (zigt) and its "novel and valuable" unifying analysis of rank
and entropy collapse under weaker assumptions than prior work (X3PM). They also
highlighted that "using the [Random Energy Model] in the context of signal
propagation analysis is novel and original" (X3PM) and noted that the paper
provides "precise asymptotic analysis" that is "useful to both theorists and
practitioners" (5iSN). Reviewer 1xRi also appreciated our "precise asymptotic
analysis".

We are particularly thankful to Reviewer 5iSN, who, after an in-depth exchange,
confirmed that all open points had been resolved and initially increased their
score to a 10 rating.

## Changes to the manuscript

We have uploaded a diff to facilitate easier tracking of all modifications, and
we confirm that the revisions keep the manuscript under ten pages. Below we
summarise the main changes to the manuscript that we have already implemented in
response to your comments (the manuscript has been updated here on OpenReview)

- **Structure:** we have collected our discussion of related work into a
  dedicated *Related Work* section.
- **Sequence geometry:** we expanded the discussion of the geometry at the
  embedding layer in Section 2 and clarified how it evolves in deeper layers in
  Section 3.2.
- **Section 4:** using the additional page granted for revisions, we now
  explicitly show how to analyse signal propagation through other transformer
  components like the MLP, clarifying the derivation of Algorithm 1.
- **Autoregressive Models**: in Section 5, we added a brief discussion on this
  topic, and Figure 12 in the appendix illustrates that our theory closely
  approximates the propagation of late sequence tokens.
- **Figures:** Figure 1e has been updated with a legend. We also added training
  curves (Figures 10 and 11) corresponding to the test-dynamics figures in the
  main text.

## Summary

We hope this summary helps the AC see how the discussion improved the paper. The
exchanges with Reviewer 5iSN also clarify points raised by X3PM (who indicated
they might increase their score if these weaknesses were addressed), and the new
version fully addresses the concerns of 1xRi and zigt. We believe the final
version is substantially clearer and more complete, and we are grateful to all
reviewers for their constructive engagement.

---

### Meta-Review · Area_Chair_f1VZ · 2026-01-06

**Summary:**

This paper provides an asymptotic characterization of signal propagation in Transformer (self-attention layer, skip connections, Layer Norm, MLP) and explain how different initialization scalings cause rank and entropy collapse. The reviewers have following concerns.
1. Experiments support is less convincing
2. The experiments concern only BERT-style architecture
3. The asymptotic prediction (Result 1) does not perfectly match the empirical results.

**Reviewer Concerns:**

I think the concerns are well addressed by the authors' sufficiently detailed responses.

**Reviewer Scores:**

4,6,6,8

---

### Decision · Program_Chairs · 2026-01-26

Accept (Poster)